# A 3D gene expression atlas of the floral meristem based on spatial reconstruction of single nucleus RNA sequencing data

Manuel Neumann [1,7], Xiaocai Xu [1,7], Cezary Smaczniak [1], Julia Schumacher [1], Wenhao Yan[1], Nils Blüthgen [2], Thomas Greb [3], Henrik Jönsson [4], Jan Traas [5], Kerstin Kaufmann [1] & Jose M. Muino [6✉]

Cellular heterogeneity in growth and differentiation results in organ patterning. Single-cell transcriptomics allows characterization of gene expression heterogeneity in developing organs at unprecedented resolution. However, the original physical location of the cell is lost during this methodology. To recover the original location of cells in the developing organ is essential to link gene activity with cellular identity and function in plants. Here, we propose a method to reconstruct genome-wide gene expression patterns of individual cells in a 3D flower meristem by combining single-nuclei RNA-seq with microcopy-based 3D spatial reconstruction. By this, gene expression differences among meristematic domains giving rise to different tissue and organ types can be determined. As a proof of principle, the method is used to trace the initiation of vascular identity within the floral meristem. Our work demonstrates the power of spatially reconstructed single cell transcriptome atlases to understand plant morphogenesis. The floral meristem 3D gene expression atlas can be accessed at http://threed-flower-meristem.herokuapp.com.

---

[1] Plant Cell and Molecular Biology, Humboldt-Universität zu Berlin, Institute of Biology, Berlin, Germany. [2] Institute of Pathology, Charité - Universitätsmedizin Berlin, Charitéplatz 1, 10117 Berlin, Germany. [3] Department of Developmental Physiology, Centre for Organismal Studies (COS), Heidelberg University, Im Neuenheimer Feld 360, 69120 Heidelberg, Germany. [4] The Sainsbury Laboratory, University of Cambridge, Bateman Street, Cambridge CB2 1LR, UK. [5] Laboratoire RDP, Université de Lyon 1, ENS-Lyon, INRAE, CNRS, UCBL, 69364 Lyon, France. [6] Systems Biology of Gene Regulation, Humboldt-Universität zu Berlin, Institute of Biology, Berlin, Germany. [7] These authors contributed equally: Manuel Neumann, Xiaocai Xu. ✉email: jose.muino@hu-berlin.de

Characterizing gene expression dynamics and heterogeneity at single-cell resolution is essential to understanding the molecular mechanisms underlying cellular differentiation in multicellular organisms. Technologies based on cell dissociation[1–3] or nuclei isolation[4–7] combined with high-throughput transcriptome sequencing (scRNA-seq/snRNA-seq) allow for the characterization of the transcriptomes of hundreds of thousands cells at single-cell resolution. However, the physical location of these cells is lost during the experimental process. In plants and other multicellular organisms, cell fate strongly depends on its precise position within the developing organism[8]. Therefore, it is essential to characterize gene expression patterns of each cell in their native physical context to fully understand the link between gene activity and organ development.

In recent years, there has been a strong development in the field of spatial transcriptomics[9–11]. However, to date, only one study in plants has been published using an early version of the 10x Visium technology with limited cellular resolution[12]. This lack of technological adaptation of spatial transcriptomics to plants maybe because of the difficulties with the enzymatic permeabilization of the cell wall. Single-molecule FISH (smFISH) and other high-resolution FISH experiments are also rarely used in plant studies[13,14] due to the endogenous autofluorescence of many plant cells[13].

Mapping of scRNA-seq transcriptomes into a computational representation of the studied organ/structure provides an alternative method for spatial reconstruction of omics data. Two seminal papers implemented this idea by mapping scRNA-seq data to a computationally binned spatial map consisting of the expression of ~100 reference genes[15,16]. This idea, with different implementations, was successfully followed by others in diverse tissues and organisms[17–22]. New methods aim to combine scRNA-seq with high-throughput spatial transcriptome data (e.g., MERFISH, Slide-seq) that collect the expression of thousands of reference genes. They are based on the projection of the scRNA-seq and the spatial transcriptomes into a common latent space e.g., SEURAT[23], Liger[24], Harmony[25], gimVI[26], SpaGe[27]. In general, there is a tendency to develop computational methods that require a large number of reference genes, which limits these tools to organisms with extensive spatial transcriptomics resources.

In plants, spatiotemporal gene expression patterns are usually established using traditional in situ *hybridization* or by confocal microscopy of promoter fusions to fluorescent reporters. Confocal microscopy has the advantage that it can be used to reconstruct 3D structures by combining several z-stack images[28–32]. In addition, combined with live image microscopy, the temporal dynamics of gene expression and morphology development can be reconstructed[32,33]. In this way, Refahi et al.[32] combined the information on spatiotemporal expression patterns of 28 regulatory genes into 3D reconstructed *Arabidopsis* flower meristems, ranging from initiation to stages 4, 5 of flower development. These methods are limited by the low number of genes profiled per experiment. Therefore, tools to integrate scRNA-seq with expression data of defined, limited sets of 3D reference gene expression patterns need to be developed for spatial reconstruction of single-cell transcriptomes in plants.

Here, we adapted novoSpaRc[34], a methodology for spatial reconstruction of single-cell RNA-seq data, to generate a 3D single-cell transcriptome atlas of a floral meristem by integrating single-nuclei RNA-seq and a 3D reconstructed flower meristem[32]. NovoSpaRc reconstruction aims to explicitly preserve the transcriptome similarity among closely located scRNA-seq cells in the spatial map, while maximizing the transcriptome similarity between the scRNA-seq cells and the cells of the spatial map to which they are assigned. In such a way, novoSpaRc performance is less affected by the number of reference genes than other

methods, and, in theory, it can also be used without any reference genes[34]. Such property makes novoSpaRc an ideal method for plant single-cell data considering the low number of available reference genes in plant tissues. However, novoSpaRc was developed to make use of spatial 2D continuous reference gene expression maps, while the 3D expression spatial map of floral meristem generated by Refahi et al.[32] is binary. We adapted the methodology for reconstructing single-cell transcriptomes in 3D making use of binary reference gene expression data. By this, we were able to generate an atlas of gene expression in different meristematic domains and spatially trace the earliest stages of tissue differentiation within the *Arabidopsis* flower. In summary, these results provide a primer for future initiatives to generate plant organ 3D atlases of gene expression.

## Results

**snRNA-seq of *Arabidopsis* floral meristems**. In order to obtain genome-wide gene expression profiles in the floral meristem at the single-cell level, we use a system for synchronized floral induction[35] (*pAP1*:AP1-GR *ap1-1 cal-1*[35]) to maximize the collection of plant material from the desired developmental stage (stage 5, 4 days after DEX-induction). We chose to study stage 5 of flower development because of the availability of several –omics datasets from this stage[35–37], which are needed to validate the performance of the method. At stage 4, 5[38], the flower whorls get established by homeotic gene activity, therefore being an excellent stage to study the initial steps of floral organ specification.

We performed single-nuclei RNA-seq (snRNA-seq)[4,7], where nuclei were collected by fluorescence-activated DAPI-stained nuclei sorting (FANS) after 4 days (stage 5) of DEX-induction. We isolate nuclei instead of protoplasts to avoid the transcriptome changes that protoplast may create[4,39,40]. After, snRNA-seq datasets were produced using the 10x Chromium system. In this way, Cell Ranger v3.1.0 identified 7716 single-nuclei transcriptomes with a median of 1110 genes expressed per nucleus. The low number of reads mapping to mitochondria genes (<5%) indicates low organelle contamination (Supplementary Fig. 1). Figure 1a shows that snRNA-seq is able to recapitulate (R = 0.88) the expression profile of bulk RNA-seq data obtained from the same stage and tissue type. Analysis of the data using Seurat v3.2.3 identified 12 main clusters and the marker genes defining these clusters (Supplementary Data 1). To annotate the clusters, we identified the top 20 marker genes specific for each cluster and plotted the expression of these marker genes in publicly available bulk RNA-seq datasets of different tissues and floral stages (Fig. 1d and Supplementary Fig. 2). In addition, we calculated the average expression of known floral meristem marker genes in the 12 snRNA-seq clusters (Fig. 1c).

We were able to recover the main tissue types present in the meristem, including different epidermal as well as vascular tissue types. The clusters appear to be dominantly grouped by the tissue where they are located (epidermis versus vasculature, and parenchyma), and their cell cycle status. The four epidermis clusters (0, 9, 10, and 11) show specific expression of *MERISTEM LAYER 1* (*ATML1*[41] and *PROTODERMAL FACTOR 1/2* (*PDF1/2*)[42] (Supplementary Data 1 and Supplementary Fig. 3). Clusters 0 and 9 are distinguished by the expression of individual marker genes such as *TRIPTYCHON (TRY)*[43], *TRICHOMELESS1 (TCL1)*[44], and genes involved in wax composition which indicates epidermal cells that will develop trichomes (cluster 0) or not (cluster 9). Clusters 10 and 11 represent dividing epidermal cells, marked by the expression of genes coding for histones which is characteristic of the S-phase and genes involved in cell division (Supplementary Data 1 and Supplementary Fig. 3).

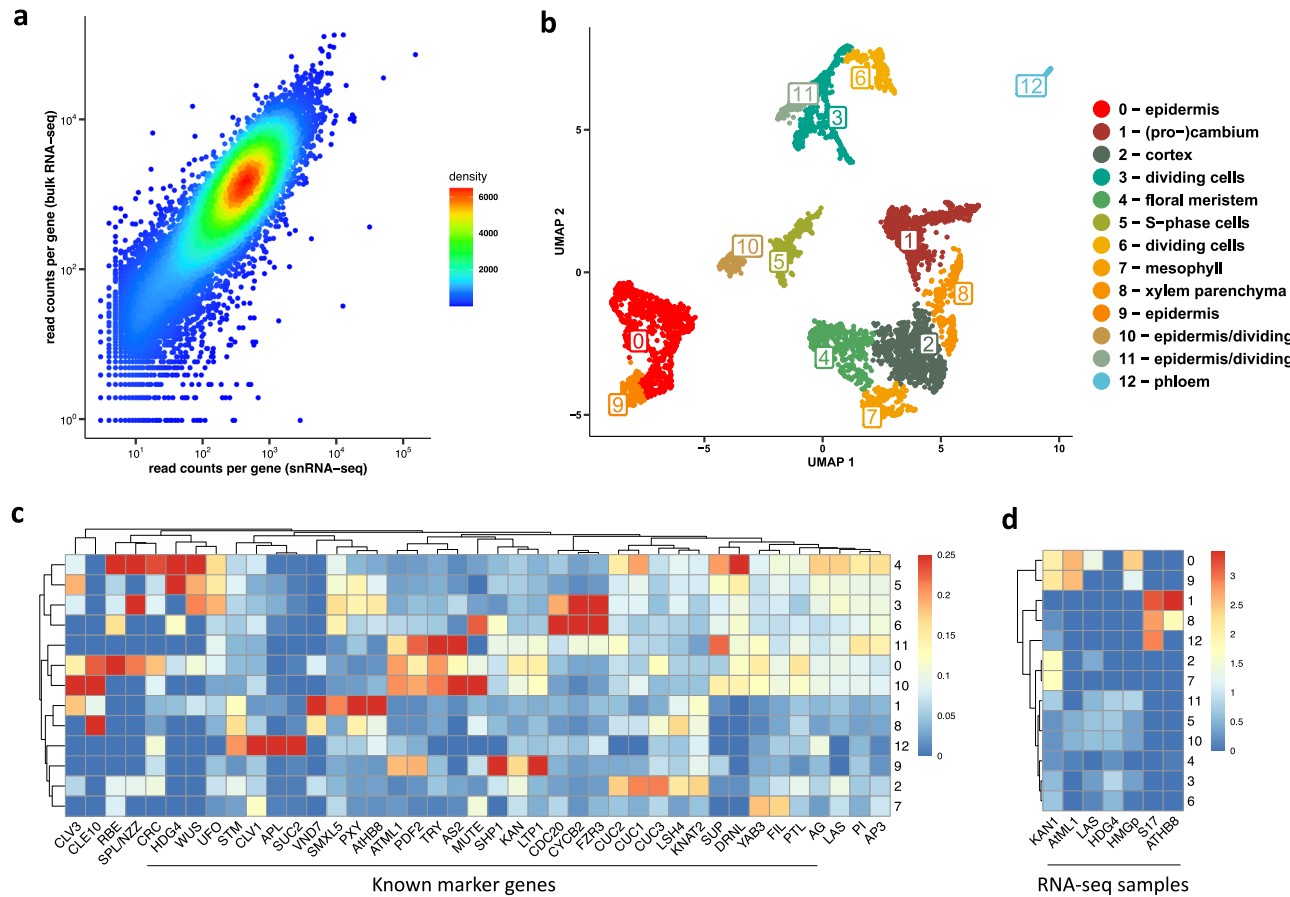

**Fig. 1 Single-nucleus RNA-sequencing of *Arabidopsis* floral meristems. a** Reproducibility (R = 0.88) of the gene expression estimated from computationally pooling all nuclei from our snRNA-seq compared to bulk RNA-seq of stage 5 flower meristem (average of three biological replicates). **b** UMAP plot and clustering snRNA-seq analysis of *Arabidopsis* floral meristems obtained by Seurat analysis. **c** Average relative expression of known floral markers on the identified snRNA-seq clusters. **d** Relationship between domain-specific shoot apical meristem bulk RNA-seq datasets profiled by (Tian et al.[67]) and the snRNA-seq clusters. The heatmap shows the relative average expression of the top 20 marker genes for each snRNA-seq cluster (y-axis) on domain-specific shoot apical meristem bulk RNA-seq datasets. For example, the top 20 marker genes of cluster 1 have a high specific expression on the ATHB8-domain, meaning that they are specific to this domain. See Supplementary Fig. 2 for expression profiles in other plant domains/stages.

Clusters 1, 8, and 12 can be classified as vasculature (Fig. 1d). More specifically, cluster 1 corresponds to vascular stem cells, as marked by cambium (Supplementary Fig. 2c) expressing markers genes such as *PHLOEM INTERCALATED WITH XYLEM (PXY)* and *SMAX1-LIKE 5 (SMXL5)*[45] (Supplementary Fig. 3). Cluster 12 contains cells that are associated with phloem, containing the marker genes *ALTERED PHLOEM DEVELOPMENT (APL)*[45,46] (Supplementary Fig. 3). Cluster 8 is enriched for vascular xylem parenchyma genes, for example, *CYTOCHROME P450, FAMILY 708 (CYP708A3)*[45] (Supplementary Fig. 3), and shows signatures of cell expansion and cellulose biosynthesis. It should be noted that in this dataset, no mature xylem vessels or phloem sieve elements can be expected because these structures lack a nucleus.

The analysis of marker genes of cluster 2 shows an enrichment on genes involved in the starch catabolic process as well as genes expressed in the cortex such as *CHALLAH (CHAL)*[47]; (Supplementary Fig. 3) and *JACKDAW (JKD)*[48], which indicates that cluster 2 represents cortex. Cluster 4 represents the floral meristem, containing specific markers such as *APETALA3 (AP3)*[49], *REPRODUCTIVE MERISTEM 34 (REM34)*[50] (Supplementary Fig. 3). Cluster 7 corresponds to cells that differentiate into mesophyll, e.g., in sepals or pedicel, and it shows a specific expression of marker genes such as *LIPOXYGENASE 2 (LOX2)*[51] (Supplementary Fig. 3) and *REDUCED CHLOROPLAST COVERAGE (REC1)*[52].

Clusters 3, 5, 6, 10, and 11 denote dividing cells (Supplementary Fig. 3). Cluster 3 is a cluster showing enriched expression of several cell-cycle associated genes. Cluster 5 shows specific activation of many histone genes whose activity is associated with the S-phase of the cell cycle, as well as some genes involved in cell proliferation and cell growth (e.g., *AINTEGUMENTA*[53]. Cluster 6 is enriched in cell cycle markers, in particular *CELL DIVISION CYCLE 20.2 (CDC20.2)*, which accumulates in the nucleus from prophase until cytokinesis[54]. Clusters 10 and 11 are epidermal cells in different cell cycle phases as described before.

Unsupervised clustering has been successfully used for the analysis of scRNA-seq data, however, one of the major drawbacks of this approach is that it identifies groups of cells depending on their transcriptome variance, and therefore it may miss cell types of biological interest without sufficient biological variance in the system. For example, we were not able to distinguish clusters representing individual floral whorls, likely because the transcriptome variance between tissue types such as epidermis and vasculature is much greater than between different whorls, at least at this stage of development. In addition, the correspondence of each cell cluster to a particular homogeneous physiological cell type is not guaranteed. For example, cluster 1 represents vascular (pro)cambium, but close inspection of this cluster (Supplementary Fig. 3) reveals specific expression of *PXY* (a marker of proximal cambium) and *SMXL5* (a marker of distal cambium) in

separate regions of the cluster. This provides additional justification for the development of a method to map the snRNA-seq transcriptomes to a physical representation of the plant tissue/organ at study. In the next sections, we describe how we map snRNA-seq data to a spatial expression map of the floral meristem that will enable the selection of the group of cells of interest (e.g., floral whorls).

**Reconstructing gene expression by snRNA-seq and microscopy image integration**. We used novoSpaRc[34] to integrate snRNA-seq data and a published 3D reconstructed *Arabidopsis* stage 4, 5 floral meristem ("spatial map") that has information on the expression pattern of 28 genes ("reference genes")[32]. To adapt novoSpaRc to map single-nuclei transcriptomes to the 3D floral meristem map with a binary expression of the reference genes, we implemented three main modifications:

1. Filtering: snRNA-seq was performed on the "cauliflower-like" meristem plant material, which may contain cells from regions (e.g., short pedicels and stems) that were not present in our spatial map. Therefore, we set up a filtering procedure to eliminate snRNA-seq transcriptomes that were too dissimilar to the transcriptomes of the spatial map (see Material and Methods for details).

2. Genes used for calculating snRNA-seq transcriptome distances: The original novoSpaRc pipeline calculates the distance between snRNA-seq transcriptomes using a set of genes selected depending on their variability across the snRNA-seq transcriptome (highly variable genes). Because in our dataset these highly variable genes were not enriched among the known flower marker genes, we also used the top 100 genes with the highest expression correlation with the reference genes, which included very well-known floral regulator genes, to calculate this distance.

3. The distance used to calculate dissimilarity between spatial map and snRNA-seq transcriptomes: The original novoSpaRc pipeline calculates distances between transcriptomes from the spatial map and snRNA-seq data using the Euclidean distance. Because our spatial map data is binary, we also employed two other distances commonly used for binary data: Hamming and Jaccard distances.

Subsequently, we studied the performance of these modifications by calculating the area under the receiver operating characteristic (AUROC) for predicting the expression of each reference gene when this gene was removed from the spatial map during the data integration step. Supplementary Fig. 4 shows the general good performance (AUROC) of our method for each gene and parameter combination tested. Three genes, *HISTIDINE PHOSPHOTRANSFER PROTEIN 6* (*AHP6*), *AUXIN RESPONSE TRANSCRIPTION FACTOR 3* (*ARF3, ETTIN*), and *CLAVATA3* (*CLV3*), had very low performance independently of the parameters used (see next paragraph for an explanation). Therefore, we calculated the overall performance of the method as the average AUROC of all genes except *AHP6, ETTIN, CLV3*, and *WUSCHEL (WUS). WUS* was excluded due to the low number of cells ($n = 8$) where it was expressed in the spatial map. In general, modifications improved the performance of the original novoSpaRc pipeline (Supplementary Fig. 5). In particular, using the Jaccard distance had a positive impact on the performance of the method in this particular dataset (Supplementary Fig. 5). In our hands, other datasets show different optimal parameter settings, but filtering always improves the performance. For visual comparison, Fig. 2 shows the reconstructed expression of representative genes whether our modifications are applied or not. In particular, *APETALA3* (*AP3*) and

*SEPALATA3* (*SEP3*) are the genes showing the biggest differences (see also Supplementary Fig. 4). For the final prediction, modifications and the parameter values which maximized the average AUROC were used to reconstruct gene expression using the whole spatial map dataset (see Material and Methods).

As mentioned before, three genes (*ETTIN, AHP6*, and *CLV3*) had low performance (AUROC close to 0.5) for any set of parameter values used when these genes were removed from the spatial map during the data integration step. We hypothesized that this is because cells expressing these genes are not expressing any of the other reference genes used, and therefore, these cells cannot be correctly mapped. We measured this expression-dependency as the maximum Spearman correlation value of a particular gene against any other gene from the reference list in the snRNA-seq data. We call this value the predicted estimation performance (PEP) for a particular gene. Indeed, there is a strong correlation between the performance of the method (AUROC) and PEP for each gene (Supplementary Fig. 6a), which indicates that we can use it as a predictor of the performance of the method for each particular gene. As we sequentially eliminate genes from the spatial map prior to gene expression reconstruction, starting with the highest correlated reference gene, and therefore decreasing the PEP value of that reconstructed gene, we see a drop in the performance (AUROC) (Supplementary Fig. 6b). However, when we sequentially eliminate reference genes starting with the lowest correlated reference gene, there is no evident decrease in performance (Supplementary Fig. 6c).

Based on Supplementary Fig. 6a, we chose a PEP threshold of 0.13 to decide which genes ($n = 1306$) we consider to have a reliable expression prediction. We obtained this threshold as the point in Supplementary Fig. 6a where the AUCROC starts to be bigger than 0.5. As the PEP value is estimated without using the spatial map, it can be used to select a set of reference genes for future experiments in order to maximize the number of correctly predicted genes. The number of genes with high PEP values ($n = 1306$ for PEP > 0.13) is mainly influenced by the number of reference genes in the spatial map. Therefore, when using a higher number of reference genes, higher PEP values are obtained per gene (Supplementary Fig. 7).

To validate the predictions of spatial gene activity in the floral meristem, we analyzed expression patterns of a set of genes by reporter gene analysis *in planta* (Fig. 3). In brief, promoter-GFP fusions were stably expressed in *A. thaliana*, and stage 4, 5 floral meristems were analyzed using confocal laser scanning microscopy. As expected, in vivo expression patterns highly correlated with reconstructed expression patterns of genes used as reference genes (*ETTIN; SHOOT MERISTEMLESS, STM*, and *MERISTEM LAYER 1, ATML1*) as well as genes with high PEP scores, e.g., AT1G62500 (*CO2*, PEP = 0.17), while there was lower overlap with reconstructed expression patterns of genes with low PEP scores, such as *SHORT ROOT* (*SHR*, PEP = 0.15), and *PIN-FORMED 1* (*PIN1*, PEP = 0.14). In general, the prediction broadly recovered the cells and tissues that show activities of the genes, but some gene expression patterns were more restricted in the reporter gene analyses (e.g., *SHR, PIN1*). This could be explained by the limited set of reference genes that was used for the prediction, in particular in the periphery where few reference genes were available, but also by the possibility that the reporter gene constructs do not contain all regulatory elements needed for correct spatial expression of the genes.

**Identifying floral meristem expression domains and their temporal dynamics**. Another advantage of our method is that we can identify expression domains in the flower meristem. For this, we set our clustering method (see Methods) to identify 15

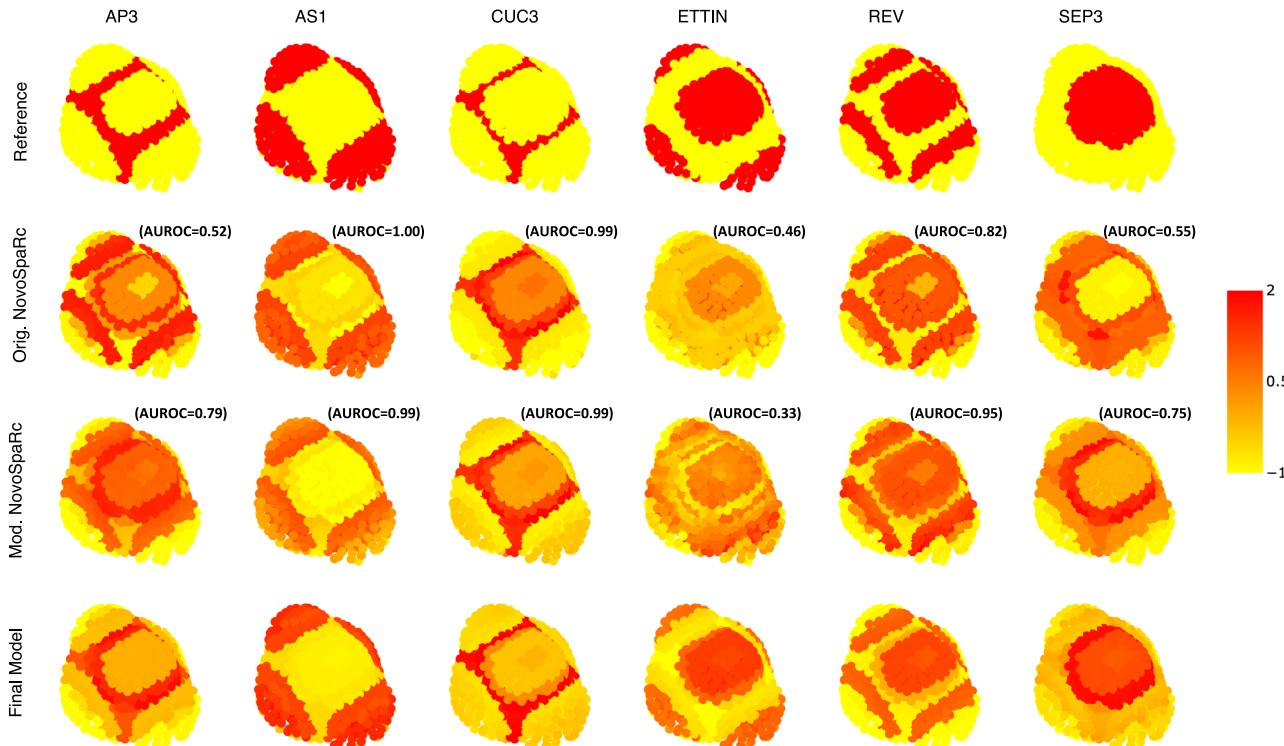

**Fig. 2 Examples of reconstructed expression patterns for representative genes in *Arabidopsis* floral meristem.** The top row shows the reference expression obtained from the spatial reference map. The second and third row is the reconstructed expression using the parameters that maximize the average AUROC when the gene to be predicted is removed from the data integration step and the original novoSpaRc pipeline (second row) or our modified pipeline (third row) is used. The bottom row is the final reconstructed expression using all the spatial map data. To facilitate visual comparison, we standardized the expression of each gene to have a mean of 0 and a variance of 1. The expression of other genes can be visualized at http://threed-flower-meristem.herokuapp.com.

expression domains in the whole flower meristem (Supplementary Fig. 8a, b). Recently, Refahi et al.[32] have identified 11 expression domains in the L1 layer of the flower meristem using a large set of RNA in situ hybridization or confocal microscopy experiments, which we reproduce in Supplementary Fig. 8c. Our method identified seven expression domains in the L1 layer (Supplementary Fig. 8a) with an almost perfect correspondence with Refahi et al.[32] (Supplementary Fig. 8d), therefore validating our methodology. We used this correspondence to transfer the names of the expression domains of Refahi et al.[32] to our identified domains in the L1 layer. In addition to the L1 layer's domains, our method is also able to identify different expression domains deep in the flower meristem which mostly are associated with the expression of vascular-specific marker genes (Supplementary Fig. 8e). The experimental validation of these new domains in the meristem is presented in further sections of this manuscript. Refahi et al.[32] may have missed these new domains because their experiment started with a pre-selection of genes that didn't include any vascular marker gene; this drawback is not present in our method since we don't need to pre-select genes to be studied. However, our method is not able to resolve the AGAMOUS domain with the resolution of Refahi et al.[32] showing that selecting genes to be used in the clustering process also has some advantages. When compared to the clusters identified in the analysis of the snRNA-seq data (Fig. 1b), our method is able to identify particular morphological domains, whereas the cluster analysis of the snRNA-seq data is limited to groups of cells with similar transcriptomes (e.g. similar cell cycle stage) that are not necessarily linked to specific flower meristem domains.

Next, to also integrate temporal information, we exemplify our methodology with a second time point. We collect plant material

from day 3 after DEX-induction using the same system for synchronized floral induction (*pAP1*:AP1-GR *ap1-1 cal-1*) as before. We chose this time point because the flower morphology is almost identical to day 4 after DEX-induction, which will allow us to use the same reference spatial map as before and, therefore, to allow us a straightforward comparison of the changes in gene expression happening between these two time points. It is important to note that no automatic method is available to link 3D microscopy-based reconstructed meristems with very different morphologies[32], so using an earlier time point, although biologically interesting, will not allow us a straightforward comparison between time points. In this way, Cell Ranger v3.1.0 identified 9792 single-nuclei transcriptomes with a median of 954 genes expressed per nucleus and a low number of reads mapping to the organelles (Supplementary Fig. 9a–d) in the sample collected three days after DEX-induction. Supplementary Fig. 9e shows that snRNA-seq is able to recapitulate (R = 0.88) the expression profile of bulk RNA-seq data obtained 4 days after DEX-induction. Using Seurat v3.2.3, we identified 11 clusters (Supplementary Fig. 9f). We integrated the snRNA-seq data with the reference spatial map using novoSpaRc with the same parameters as before. As expected, the expression domains obtained by clustering the predicted spatial expression profiles are almost identical between both time points, with just some differences in the sepal primordia (Supplementary Fig. 8a, b and Supplementary Fig. 10a, b). As we have mapped both time points to the same reference spatial map, we can calculate the average gene expression for each identified expression domain at each time point (Supplementary Fig. 10c). For example, for cluster 15:"carpel boundary" (Supplementary Fig. 10d), we identified several genes up- or down- regulated in the early time point,

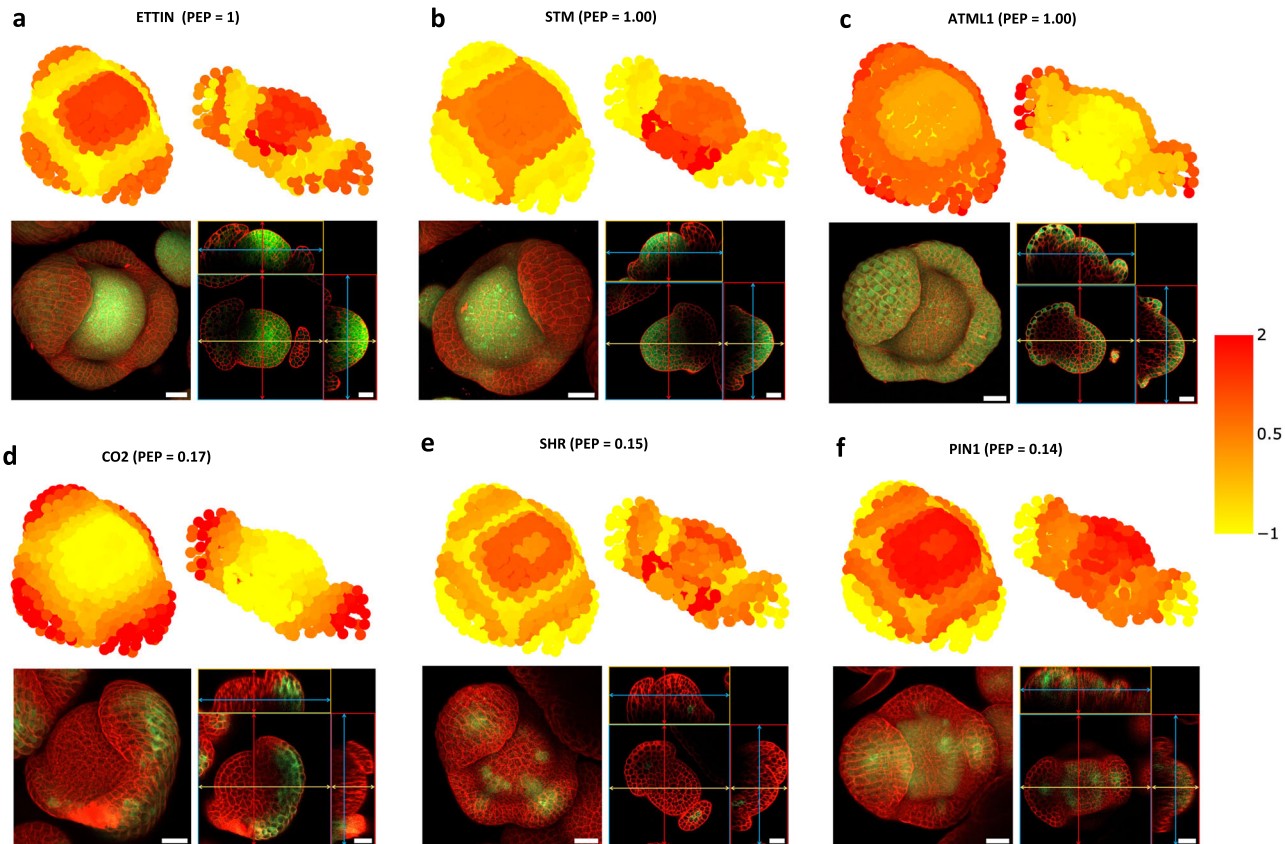

**Fig. 3 Validation of reconstructed gene expression patterns with reporter gene assays.** Upper part in **a–f** shows the predicted expression of ETTIN, STM, ATML1, CO2, SHR, and PIN1 from the top and cross-section view of stage 4–5 flower meristems. To facilitate visual comparison, we standardized the expression of each gene to have a mean of 0 and a variance of 1. Titles include gene symbol and PEP score for the predicted 3D expression profile. The lower part in **a–f** shows the GFP expression pattern (green) for plant lines under the control of the respective promoter, as detected by confocal laser scanning microscopy in *A. thaliana* stage 4–5 flower meristems. Confocal images show the flower meristem from the top (left) as well as different orthogonal sections (right). Cell walls were stained using propidium iodide (red). Scale bars indicate 20 μm. At least five independent transgenic lines were investigated for each of the ETT, STM, and ATML1promoter-reporter fusions. One single CO2, SHR, and PIN1 promoter-reporter line was obtained as part of the BREAK line set (Marquès-Bueno et al.[62]). For each of the BREAK lines, the experiment was performed at least twice.

including important floral regulators as *GENERAL REGULA-TORY FACTOR 2* (*GRF2*), *PSEUDO-RESPONSE REGULATOR 5* (*PRR5*), *D6 PROTEIN KINASE* (*D6PK*), or *LIGHT SENSITIVE HYPOCOTYLS 4* (*LSH4*). Next, we were curious if the genes that increase in expression between day 4 compared to day 3 (log2foldchange >1) for each expression domain will be good predictors of the genes expressed in the mature flower organs that originated from each of these domains. Indeed, when we plot the average expression of the set of genes increasing in each domain, we see a specificity for genes expressed in the mature floral organ that will be raised from these domains (Supplementary Fig. 10e), indicating that some of the gene programs that will be active in the mature floral organ start to be activated at this early stage of flower meristem development.

**Quantitative gene expression reconstruction of AG- and AP3-floral meristem domains.** Next, we evaluated the performance of spatial expression reconstruction on day 4 after DEX-induction to study quantitative gene expression in particular domains that give rise to the different organ types in the flower. In *Arabidopsis* flower development, the identities of different organ types are determined by floral homeotic transcription factors. In particular, sepals are specified by the activity of *APETALA1* (*AP1*), petals are defined by the combination of *AP1* and *APETALA3* (*AP3*),

stamens are specified by *AP3* and *AGAMOUS* (*AG*), and the carpels is determined by *AG* activity.

We estimated the expression of a gene in the *AP3*- and *AG*-domains of the 3D reconstructed meristem, as the average expression of that gene in the cells which had a positive expression of *AP3* or *AG* reference genes, respectively. To validate these results we generated sorted nuclei RNA-seq (FANS RNA-seq) from floral meristems expressing nuclear targeting fusion protein (NTF)[55] in *AP3* vs. *AG*-expression domains after 4 days of DEX- induction. The GFP-containing NTF protein was transcribed under the control of the *AP3* promoter (*pAP3*::NTF) and the second intron of *AG* (*pAGi*::NTF) in the floral induction system (*pAP1*:AP1-GR *ap1-1 cal-1*)[35]. The expression patterns of *pAP3*::NTF and *pAGi*::NTF were visualized by confocal micro-scopy (Supplementary Fig. 11) and the nuclei of *AP3*- or *AG*-expression domains were sorted based on the positive GFP signal in FANS.

Transcriptomes retrieved from the spatially reconstructed *AP3* and *AG* domains in the floral meristem showed a high correlation with the domain-specific bulk RNA-seq expression (Rho = 0.89 for *AP3*- and Rho = 0.88 for *AG*- domain when using genes with a PEP higher than 0.13). This was close to the correlation obtained among the bulk RNA-seq biological replicates (Rho = 0.95 for *AP3* and Rho=0.93 for *AG*) when using the same set of genes (Fig. 4), which indicates a very good performance of the

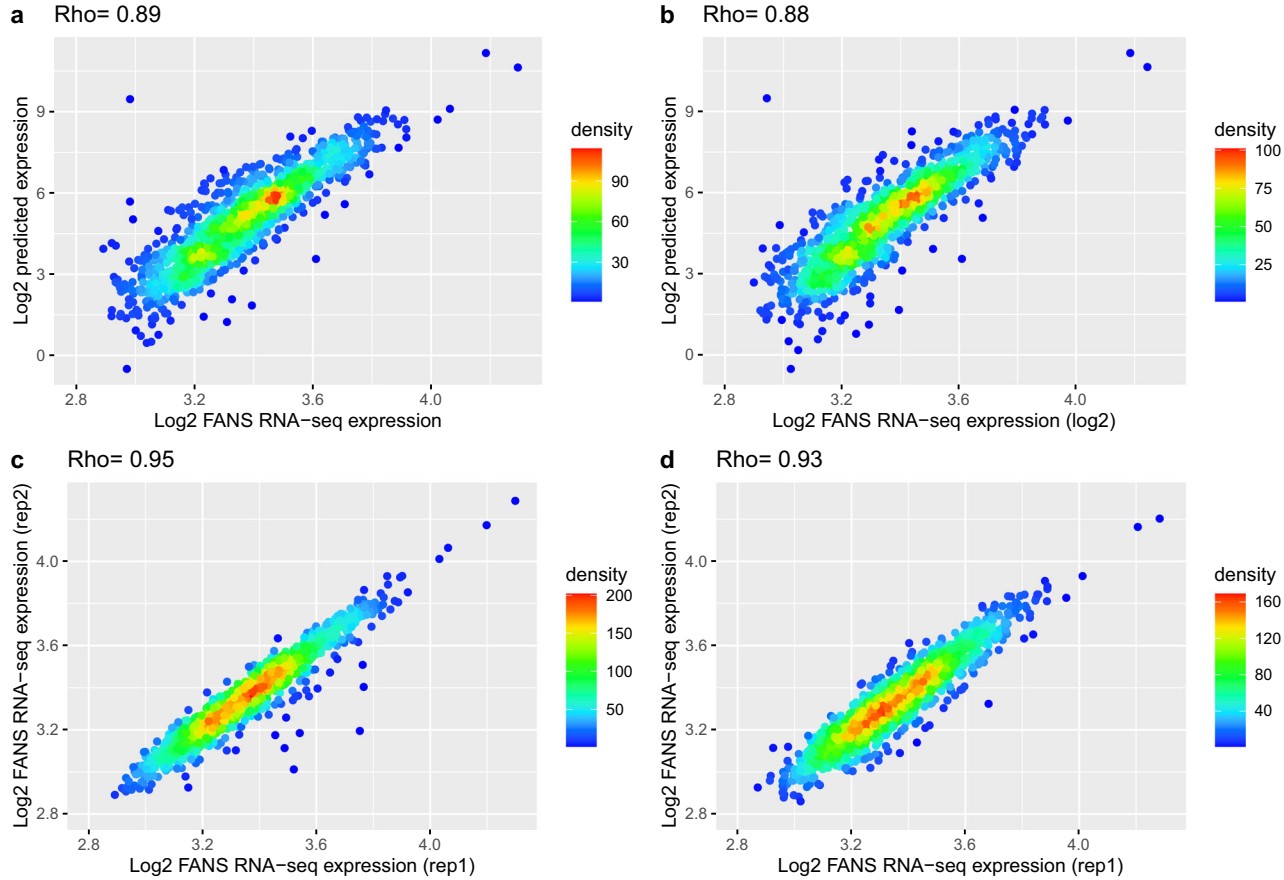

**Fig. 4 Prediction of AP3 and AG domain gene expression.** Scatterplot showing the gene expression for AP3 (**a**) and AG (**b**) domain predicted by our method (y-axis) and observed by our FANS bulk RNA-seq data (x-axis) when using genes with PEP value >0.13 ($n = 1306$). The bottom row shows the scatterplot for the gene expression of both biological FANS bulk RNA-seq replicates for AP3 (**c**) and AG (**d**). On the top of each figure, the Spearman's rank correlation coefficient (Rho) is reported.

method. Even more interesting, the reconstructed expression was able to recover the $\log_2$ fold-change gene expression between both domains (Supplementary Fig. 12a, Rho = 0.37) when using genes with a PEP higher than 0.13 ($n = 1306$). In particular, the obtained correlation was very close to the correlation of the $\log_2$ fold-change gene expression obtained from the bulk RNA-seq biological replicates when using the same set of genes (Rho = 0.47, Supplementary Fig. 12c). This indicates that spatially reconstructed transcriptomes are able to predict domain-specific differential gene expression. The correlation between gene expression prediction and domain-specific bulk RNA-seq increases with increasing PEP scores (Supplementary Fig. 12b), which is in agreement with the notion of the PEP score being an indicator of the quality of the predicted 3D expression.

In this way, we detected a large number of genes with specific expression (Supplementary Data 2) in the different floral whorls as determined by the (combined) expression of *AP1* (sepal), *AP3-AP1* (petal), *AP3-AG* (stamen), and *AG* (carpel). For example, we predict a higher expression of *APETALA2* in the sepal domain, which is in line with its known role in sepal specification together with *AP1*[56]. We predicted *PETAL LOSS* (*PTL*) expression in the *AP1* and *AP1-AP3* domain, which is consistent with previous findings that *PTL* is expressed in sepal margins while controlling petal development[57]. On the other hand, we predict *PERI-ANTHIA* to be strongly induced in the three inner whorls, as expected from the literature[58], while we predicted *UNUSUAL FLORAL ORGANS* (*UFO*) to be expressed in the *AP3-AG* and *AP3-AP1* domain which fits with the observed expression in the

petal and stamen whorls[59]. This exemplifies the power of the method to identify whorl-specific genes. The predicted floral whorl-specific expression is significantly related to the direct DNA-binding of flower domain-specific TFs in their regulatory regions (Supplementary Data 2 and Supplementary Fig. 13).

It is worth noting that we could apply a similar methodology directly to the snRNA-seq data (w/o 3D reconstruction), where average domain-specific expression is calculated as the average expression among the snRNA-seq transcriptomes of nuclei that have a positive expression of *AP3*, or *AG* for each domain respectively. However, the obtained fold-change expression has low agreement with the domain-specific bulk RNA-seq data (Rho = 0.04 pv < 0.14, Supplementary Fig. 14) when using the same genes as before (PEP > 0.13). This indicates that the advantage of integrating the 4395 transcriptomes of the snRNA-seq data into a physical map of 1331 cells has the additional benefit of obtaining a more accurate estimate of gene expression per cell as it is calculated since it combines the information from several snRNA-seq transcriptomes.

In summary, the presented data demonstrates that our method can be used to create a genome-wide 3D gene expression atlas of a plant organ, and to correctly predict gene expression and gene fold-change expression of particular morphological regions that was not possible with the snRNA-seq data alone.

**The origin of vascular cell identity in the floral meristem.** Spatial reconstruction of transcriptomics data can be used to pinpoint the spatial location of cells characterized with a particular transcriptome signature (e.g., snRNA-seq cell clusters,

ploidy levels[60], vascular cells[45]) by using an expression similarity-based method. For example, the initial establishment of vascular stem cell identity in the apical meristems is not well known[61]. The transcriptomes of vascular tissues in inflorescence stems have been characterized by FANS bulk RNA-seq[45], including *SMXL5* (distal cambium) and *PXY* (proximal cambium). Therefore, assuming that the vascular tissues have similar transcriptomes in the inflorescence stem and in the floral meristems, we could predict the location of vascular stem cells on the reconstructed 3D meristem even when they cannot be distinguished anatomically. We indeed obtained a distinct distribution pattern of vascular stem cells (Supplementary Fig. 15), where the cambium (*PXY* and *SMXL5*) localizes in the cell layers adjacent and just below the floral meristem with a radial disposition. Confocal imaging confirmed that *PXY* and *SMXL5* expression is initiated in cells just adjacent/below the apical meristem, but in a specific subset of cells along the central axis of the meristem (Supplementary Fig. 15). This discrepancy could be due to the low number of reference genes used, which may not allow having the needed resolution. Once these cells have been located, their transcriptome can be estimated as explained before, obtaining a good correlation (Rho = 0.34–0.42; Supplementary Fig. 16) when compared with the FANS bulk RNA-seq data. This information can be used in future work to characterize the molecular control and regulatory networks of initiation of vascular identity in the floral meristem.

NovoSpaRc outputs the probability of each snRNA-seq transcriptome as corresponding to a particular cell in the spatial map. Therefore, we can map the location of the identified snRNA-seq clusters (https://threed-flower-meristem.herokuapp.com) and visualize their physical location. In particular, cluster 1-(pro) cambium shows the same location adjacent/below the apical meristem as the one estimated by transcriptome similarity.

In summary, this shows the potential to integrate different features (e.g., cells differentiating to vascular tissues) into a common spatial map which can be used to associate with the spatial expression profiles.

## Discussion

The identity and function of plant cells are strongly influenced by their precise location within the plant body[8]. Therefore, to understand plant development at the molecular level, it is important not only to characterize the molecular status and dynamics of each individual cell but also to know their physical location in the plant. As stated in the introduction, spatial genomics in plants has been limited to profiling only a limited number of genes per experiment. Here, we provide a proof of concept for a methodology to overcome this limitation by combining scRNA-seq/snRNA-seq with a 3D microscope-based reconstructed floral meristem. The core of our approach is based on the use of the tool novoSpaRc to map the scRNA-seq data to the spatial map. In contrast to other methods that were designed to use hundreds[15,16] or thousands of reference genes[27] to map both datasets, novoSpaRc requires very few reference genes. For example, in this work, we successfully used it using just 23 reference genes. Therefore, meanwhile experimental methods to profile the spatial expression pattern of a large (>100) number of genes are not available in the plant field, we believe that the novoSpaRc method should be the preferred option to integrate scRNA-seq and spatial maps in the flower meristem. Recently, Bravo González-Blas et al.[31] implemented a new methodology to map scRNA-seq and/or scATAC-seq to a 1D or 2D spatial map. The method doesn't require any reference gene and it is based on: (1) Linking the known expression domains in the spatial map to the cell clusters identified and annotated in the scRNA-seq (e.g., cells from the cluster annotated to epidermis will be linked to the

epidermis regions of the spatial map). (2) To use pseudo-time analysis to order the transcriptomes of the scRNA-seq in a spatial axis, this assumes that there is only one axis of variation in the spatial expression patterns (e.g., ventral-dorsal axis), and the expression patterns are symmetric around the axis. Although, we can imagine that this will be a valid approach to some root tissues, where one main axis of expression variation is present (from root cap to maturation region), it is not the case for the complex 3D morphology of the flower meristem.

Using NovoSpaRc, we are able to reconstruct the spatial expression of a large number of genes (>1000) in their native spatial context. Moreover, we were able to quantitatively estimate the expression of these genes in particular morphological regions of the floral meristem. Future work should develop more dedicated statistical methods to test for gene expression differences on the 3D reconstructed structure. One possibility is to apply a re-sampling approach to the snRNA-seq data. We envision that by independently mapping multiple subsamples of the snRNA-seq data to the reference map, we will be able to estimate the variance of the gene expression which is needed to test for differential gene expression in different (groups of) cells.

The number of high-quality genes predicted is heavily dependent on the number and identity of genes present in the reference spatial map. Thus, we provide a PEP score that can be used to estimate the performance of the predicted expression for each gene, even before having generated the reference spatial map. In this way, this score can be used to select the minimum set of reference genes needed to obtain a good prediction of the spatial expression of a desired group of genes. Hence, this score helps in planning the design of a spatial genomics experiment whose data will be used as a spatial reference to predict the spatial expression of a set of genes.

This methodology has the potential to be applied to other types of -omics experiments. For example, it could be applied to map scATAC-seq experiments into the 3D reconstructed floral meristem. This offers the additional benefit of being able to integrate multiple single cells -omics data in their natural physical context. Indeed, an important problem is how to integrate multiple single-cell-omics experiments (e.g., scRNA-seq and scATAC-seq data). The typical approach[23] is to find anchors between genes and ATAC-seq regions that allow us to link the cells profiled independently in both types of experiments. We envision that independently mapping the scRNA-seq data and the scATAC-seq to a common spatial map will be an alternative way to integrate both types of experiments. In addition, we have shown that we can map transcriptional signatures of particular features (e.g., cells differentiating from vascular tissues) to the reconstructed spatial map, allowing us to annotate or integrate additional experiments/data in the spatial map.

Furthermore, time-series scRNA-seq datasets could also be tackled with this approach. For example, when live imaging has been used to reconstruct the spatial map at different time points and cell segmentation and lineage tracking have been used to infer cell lineage in the spatial map[32], the inferred cell lineage can be used to link the cells at different time points. Alternatively, when plant structure at the different time points considered has similar morphology, the scRNA-seq data could be mapped to the spatial map of one particular time point. Otherwise, computational alignment of the spatial maps at each time point will be required.

In summary, these results provide a primer for future initiatives to generate plant organ 3D atlases and for studies aiming to understand single-cell-omics studies with regard to plant morphology and development.

## Methods

**Plant material**. *pAP1*:AP1-GR *ap1*-1 *cal*-1 plants[35] were grown at 22 °C under long-day conditions (16 h light, 8 h dark) on the soil. After plants bolted and

reached the height of 2 to 5 cm, they were induced daily by applying the DEX-induction solution (2 µM Dexamethasone and 0.00016% Silwet L-77) to their main inflorescences. Around 20 inflorescences were collected and snap-frozen in liquid nitrogen on day 3 and day 4 after the first DEX-induction for snRNA-seq and on day 4 after induction for the domain-specific RNA-seq library preparation.

**Nuclei isolation.** Inflorescences were gently crushed to pieces in liquid nitrogen using a mortar and a pestle and then transferred to a gentleMACS M tube. After liquid nitrogen evaporated totally, 5 ml of Honda buffer (2.5% Ficoll 400, 5% Dextran T40, 0.4 M sucrose, 10 mM MgCl$_2$, 1 µM DTT, 0.5% Triton X-100, 1 tablet/50 ml cOmplete Protease Inhibitor Cocktail, 0.4 U/µl RiboLock, 25 mM Tris-HCl, pH 7.4) was added to the tube. Nuclei were released at 4 °C by homogenizing the tissue in an M Tube using the gentleMACS benchtop dissociator. The M Tube bears septums, a rotor, and a stator on the cap. After being attached to the gentleMACS dissociator, the M tube can homogenize plant tissues into single-nuclei suspension automatically with a defined program. Our program consists of several rounds of spin commands with different speeds (200–400 rpm) and durations and the integration of counter-clockwise rotations in between. The whole program lasts for around 5 min (see Supplementary Data 3). The resulting homogenate was filtered through a 70-µm strainer, and another 5 ml Honda buffer was applied onto the filter to collect the remaining nuclei. Nuclei were then pelleted by centrifugation at 1000 × g for 6 min at 4 °C and then resuspended gently in 500 µl Honda buffer. The nuclei suspension was filtered again through a 30-µm strainer, diluted by adding 500 µl PBS buffer and stained with 2 µM DAPI. Ambion RNase Inhibitor and SUPERaseIn RNase Inhibitor were added to a final concentration of 0.4 and 0.2 U/µl, respectively. 200,000 events of single-nuclei were selected on DAPI signals by a BD FACS Aria Fusion into a 1.5-ml tube with landing buffer (15 µl 4% BSA in PBS with 80 U Ambion RNase Inhibitor and 80 U SUPERaseIn RNase Inhibitor). Sorted nuclei were counted in Neubauer counting chambers under a Leica DMi8 fluorescent microscope.

**Preparation of snRNA-seq libraries.** Single-nuclei RNA-seq library was prepared from 10,000 freshly-isolated plant nuclei with the Chromium Single Cell 3′ Reagent Kits v3 according to the manufacturer's instructions. 14 PCR cycles were used for cDNA amplification, and 13 PCR cycles were used for the final amplification of day 4 after the DEX-induction snRNA-seq library. 11 PCR cycles were used for cDNA amplification, and 14 PCR cycles were used for the final amplification of day 3 after the DEX-induction snRNA-seq library. The average fragment size of the snRNA-seq library was checked with an Agilent High Sensitivity D1000 ScreenTape, and the concentration was measured with Qubit 1X dsDNA HS Assay Kit. Sequencing was performed on a HiSeq 4000 (Illumina) platform for day 4 after the DEX-induction sample and NovaseqS2 (Illumina) for day 3.

**Preparation of domain-specific RNA-seq libraries.** Nuclei were from pAP1:AP1-GR ap1-1 cal-1 transgenic plants expressing a GFP labeled nuclei envelope protein driven by tissue-specific promoters[55]. We used AP3 promoter and AG 2nd intron plus a minimal 35 S promoter element as promoters for the constructs to mark AP3 and AG expressing domains in flowers, respectively. After nuclei isolation, as described in the previous paragraph, nuclei were sorted into a 1.5-ml tube with 15 µl of 4% BSA in PBS and 6 µl of RiboLock RNase Inhibitor by a BD FACS Aria III. The GFP channel was set using pAP1:AP1-GR ap1-1 cal-1 as a negative control, and then nuclei were selected by gating on the DAPI peaks under the GFP positive events. After sorting, nuclei were pelleted at 1500 × g for 10 min at 4 °C, and the supernatant was then removed. Nuclei were lysed by vortex in 350 µl RLT buffer with 2-Mercaptoethanol, and RNA was then isolated with Qiagen RNeasy Micro Kit. After RNA isolation, cDNA synthesis was done with SMART-Seq® v4 Ultra® Low Input RNA Kit following the manufacturer's instructions. cDNA was sheared to 200–500 bp size by Covaris AFA system and constructed with sequencing adapters by ThruPLEX DNA-Seq Kit.

**Confocal imaging.** GFP expressing plant lines under the control of the CO$_2$(AT1G62500), PIN1 (AT1G73590 and SHR (AT4G37650) promoters were obtained from the Nottingham Arabidopsis Stock Centre (NASC, UK) as part of the SWELL line seed collection (BREAK line set N2106365), which was previously generated by Marquès-Bueno et al.[62]. To generate plant lines driving GFP expression from the ETT/ARF3 (AT2G33860) promoter, we inserted a 3 kb long promoter fragment into the pK7GW-INTACT_AT vector (Ron et al.,[63]) using gateway cloning. Similarly, the 6.1 kb promoter of STM (AT1G62360) and the 5 kb promoter of ATML1 (AT4G21750) were introduced into the pK7GW-INTACT_AT vector. A. thaliana Col-0 wild-type plants were transformed by the floral dip method (Clough and Bent, 1998). Plant lines expressing HISTONE 4 (H4)-coupled GFP under the control of the PXY (AT5G61480) and the SMXL5 (AT5G57130) promoters (PXY:H4-GFP/SMXL5:H4-GFP) were previously described by Bravo González-Blas et al.[45]. For GFP expression analysis, plants were grown on the soil at 22 °C and 16/8 h light/dark cycles using daylight led lights (200 µmol m$^{-2}$ s$^{-1}$).

GFP expression was detected by confocal laser scanning microscopy using the Zeiss LSM 800 confocal microscope equipped with a Plan-Apochromat 20×/0.8 M27 or a C-Apochromat 40×/1.2 W Korr objective. GFP was excited at a wavelength of 488 nm with an argon laser, while emission was filtered by a 410–532 nm bandpass filter. Propidium iodide (Sigma-Aldrich) was used to stain

cell walls. It was excited at a wavelength of 305 nm and detected in a range of 595–617 nm. Z-stack images were median corrected and merged to orthogonal projections using the ZEN imaging software (Zeiss).

**snRNA-seq data analysis.** Fastq files were processed with Cell Ranger v3.1.0 with default parameter values and using the Araport11 gene annotation[64], obtaining 7,716 nuclei transcriptomes as a read count matrix for day 4 after DEX-induction and 4504 nuclei transcriptomes for day 3 after DEX-induction. Genes encoded in the organelles were removed. Next, read count normalization and clustering were done with the R package Seurat v3.2.3[23]. In particular, nuclei transcriptomes with less than 1000 expressed genes were removed and SCT-normalization was applied within the SEURAT package setting the parameter *variable.features.n* to 3,000 and other parameters to default values. Next, the optimal number of PCAs was chosen to be the first nine principal components by plotting the standard deviations of the principal components using the *RunPCA* and *ElbowPlot* functions. UMAP dimensionality reduction was obtained with the *runUMAP* function using the parameters *values dims = 1:9, reduction = "pca", n.neighbors = 50, min.dist = 0.01, umap.method = "uwot", metric = "cosine"*. In order to identify clusters in the UMAP space, we used *Find-Neighbors* and *FindClusters* functions with parameter values *resolution* = 0.04, *algorithm* = 1 and default values for other parameters. Marker genes for each cluster were identified with the function *FindAllMarkers* and parameter values: *only.pos = TRUE, assay = "SCT", slot = "scale.data", min.pct = 0.25, logfc.threshold = 0.25*. In order to annotate the identified clusters, the average relative expression of the top 20 cluster marker genes in different publically available RNA-seq (see bulk RNA-seq analysis) and microarray samples were visualized in heatmaps in order to help to annotate the clusters. Expression values for GSE28109[65] were downloaded directly from the GEO omnibus (file: GSE28109_averaged_mas5_data.txt). Heatmaps showing the expression of markers genes were calculated as the average relative expression across all nuclei for each cluster. Relative expression was calculated as the normalized read count expression of a gene minus the average expression of this gene across all samples/nuclei considered.

**Bulk RNA-seq analysis.** Fastq files from publicly available bulk RNA-seq data were downloaded from Sequence Read Archive (SRA; https://www.ncbi.nlm.nih.gov/sra). The next analysis was done for each dataset independently. The analyzed datasets were: PRJNA314076[66], PRJNA471232[67]; PRJNA595605[45], and the AG- and AP3-domain-specific bulk RNA-seq data generated in this project. Fastq files were trimmed from adapters using Trimmomatic v0.36[68]. The reads were then mapped to the TAIR10 Arabidopsis genome using STAR v2.7.0b[69] with parameter values --alignIntronMax 10000 --outFilterMultimapNmax 1 --outSJfilterReads Unique and other parameters with default values. *featureCounts* v1.6.4[70] was used to count the number of mapped reads per gene (in exon and introns) with default parameters. Next, reads mapping to genes encoded in the organelles were removed. Only genes with more than ten reads mapped in at least two samples were considered in the further analyses. Read count data were analyzed with DESeq2 v1.24.0[71], in particular, normalized expression was calculated with *variance stabilizing transformation* function using default parameters.

**snRNA-seq and spatial gene expression map data integration.** snRNA-seq data were processed as described in the previous section, which results in a matrix of normalized expression values of 6104 nuclei and 19,718 genes for day 4 after DEX-induction and 4504 nuclei and 19,497 genes for day 3. Genes expressed in less than 30 cells were removed (n = 2890) with the exception of WUS and CLV3 which were kept in the dataset due their biological importance. Data of the spatial map containing positional coordinates of 1451 cells, their associated cell growth, cell volume, lineage, and expression of 28 genes for the reconstructed 3D stage four floral meristem was downloaded from Refahi et al.[32]. First, cells (n = 52) with an expression of none of the 28 reference genes were removed. Next, genes (n = 5) with the same expression in all nuclei or not present in the normalized snRNA-seq dataset were removed as they are not informative for the data integration procedure. Cells from the spatial map (n = 68) were removed when they had less than three reference genes expressed, or when the combination of genes expressed in one particular cell was present in less than four other cells. This resulted in a spatial map of 1331 cells and 23 genes. Next, nuclei from the snRNA-seq datasets not expressing any of the 23 genes considered in the reconstructed meristem were removed. At this step, the snRNA-seq contained 5910 nuclei and 16,828 genes for day 4 and 4504 nuclei and 16,496 genes for day 3. The resulting snRNA-seq dataset and the reconstructed floral meristem were integrated using novoSpaRc v0.4.1[34]. As described in the main text, three modifications were considered for day 4:

1. Filtering. When this modification was applied, distances between all the transcriptomes of the snRNA-seq and the spatial map were calculated. Only the top 50 snRNA-seq transcriptomes with the closest distance to each cell of the spatial map were kept in order to eliminate nuclei that were not present in the spatial map (e.g., cauline leaves, pedicel…). The final number of snRNA-seq nuclei depends on the distance used.

2. Genes are used for calculating distance among the snRNA-seq transcriptomes. The standard novoSpaRc procedure uses the highly variable genes identified by the program to analyze the snRNA-seq data in order to calculate the distances among the snRNA-seq transcriptomes. We modified this option to use the top 100 genes with the highest Pearson correlation

value in the snRNA-seq space to the 23 genes considered in the spatial map. In our case, this results in 1709 unique genes.

3. Distance. By default, novoSpaRc used the Euclidean distance between the snRNA-seq and spatial map transcriptomes. We also included Jaccard and Hamming distances for binary data. When these distances were used, the snRNA-seq data was binarized as non-expressed when the normalized expression of a gene was zero and as expressed when the normalized expression was bigger than zero. When using the Euclidean distance, we include the optional binarization of the snRNA-seq expression data.

The best set of modifications and parameter value sets was chosen as the ones minimizing the average AUCROC of the genes from the spatial map except *AHP6*, *ETT*, *WUS*, and *CLV3*, we excluded these four genes because their performance was always poor independently of the parameter values used and/or because the low number of cells where they were expressed in the spatial map. The final parameter set was using all three proposed modifications, in particular using the Jaccard distance, and with values for the novoSpaRc parameters: $num\_neighbors\_source = 2$, $num\_neighbors\_target = 5$, $epsilon = 0.05$, $alpha = .1$, $max\_iter = 5000$ and $tol = 1e-9$. As output, novoSpaRc provides a matrix (Gromoth-Wasserstein matrix, GW) containing the probabilistic assignment of each nucleus from the snRNA-seq to each of the cells of the spatial map. For numerical reasons (to avoid long decimals), the GW matrix was multiplied by $10^5$. It also outputs the predicted expression of each gene considered in the spatial map space. For day 3, we used the same parameters described above to allow a better comparison between time points.

**Identification of expression domains**. In order to identify expression domains in the flower meristem using our reconstructed expression profiles, we first selected high variable genes as the genes with a PEP score bigger than 0.13 and a variance bigger than 3. Next, the expression of each gene was log2-transformed and, after, standardized to mean 0 and variance 1. Then, we clustered the transcriptomes based on the euclidean distance of the previously selected high variable genes expression profiles using the hierarchical clustering algorithm implemented in the R function *hclust* with method = "average". We choose to identify 15 clusters or expression domains. For each cluster/expression domain, we calculated the expression of each gene for each domain as the average expression of those genes in all the cells of that domain. Next, we obtained the list of genes with a log2fold-change bigger than one between day 4 and day 3 after DEX-induction. For each of these list of upregulates genes, we calculated their median expression (read per million) in mature floral organs using the expression data from PRJNA314076[66]. Supplementary Fig. 10e shows relative expression calculated by subtracting the median expression of each list of upregulated genes versus the median expression of all genes whose expression is predicted with a PEP score >0.13.

**PEP score calculation**. The Spearman correlation coefficient for a particular gene against each reference gene was calculated in the scRNA-seq data after the filtering step. The highest Spearman correlation coefficient was chosen as the PEP score for that particular gene.

**Localization of the vascular stem cells into the spatial map**. FANS RNA-seq data[45] was analyzed as explained above. After, the data were log2-transformed, and the expression of each gene was normalized to have a mean of 0. The same procedure was applied to the gene expression profiles of the spatial map. Pearson correlation was calculated between each FANS RNA-seq dataset to the transcriptome of each cell of the spatial map. Only genes ($n = 1281$) defined as vascular markers in[45] were used to calculate the correlation. P-values were calculated by testing if the correlation was higher than zero.

**Localization of the snRNA-seq clusters into the spatial map**. NovoSpaRc outputs the probability of each snRNA-seq transcriptome as corresponding to a particular cell in the spatial map (GW matrix). Once obtained, the GW matrix was transformed so that columns (corresponding to cells in the spatial map) sum to 1. The score of one cell of the spatial map belonging to a particular cluster was calculated as the sum of the probabilities of all snRNA-seq transcriptomes of one particular cluster belonging to that particular cell in the spatial map.

**Reporting summary**. Further information on research design is available in the Nature Research Reporting Summary linked to this article.

## Data availability
The snRNA-seq and bulk RNA-seq data have been deposited in the GEO database under accession number GSE174599 and GSE174656, respectively.

## Code availability
Custom scripts will be provided upon request.

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

## Acknowledgements

We thank Johanna Müschner for their technical support. This work was supported by the BMBF-funded de.NBI Cloud within the German Network for Bioinformatics Infrastructure (de.NBI). The work was supported by DFG (grant no. KA 2720/5-1 to X.X., K.K., and grant KA 2720/9-1 to M.N. and K.K.), by an ERC Consolidator grant (PLANTSTEMS, #647148) to T.G. We acknowledge support by the Open Access Publication Fund of Humboldt-Universität zu Berlin.

## Author contributions

M.N., X.X., C.S., K.K., and J.M.M. jointly conceived and designed the study. M.N. and J.M.M. performed the computational analysis. X.X., C.S., and J.S. conducted the single-cell and microscopy experiments. W.Y., T.G., N.B., H.J., and J.T. provided plant lines and other resources for the study. M.N. and J.M.M. wrote the manuscript with input from all authors.

## Funding

## Competing interests

The authors declare no competing interests.
