## [Peer Review File · Nature Communications]

A 3D gene expression atlas of the floral meristem based on spatial reconstruction of single nucleus RNA sequencing dataREVIEWER COMMENTS

Reviewer #1 (Remarks to the Author):

I am generally positive on this manuscript because authors demonstrated how they used novoSpaRc to construct a 3D expression atlas of Arabidopsis floral meristem, where the transitional unsupervised clustering approach is failed due to the dominance of the epidermal/vascular tissues and dividing cells. This strategy will be useful and of broad interest to the researches working on scRNA-seq in plants. Moreover, authors have generated a webserver which facilitates the use of the datasets generated in this study. However, the paper can be improved in the following aspects:

- 1) Line 175, Authors' stated the drawback of unsupervised clustering approach. However, it should be noted that several scRNA-seq analyses in Arabidopsis roots have successfully reconstructed gene spatial expression map and inferred underlying differentiation trajectories. These papers should be cited and discussed. Please also note that Zhang et al (Dev Cell, 2021) have proposed another approach to mitigate the dominance of certain cell types during cell clustering. They were able to infer differentiation trajectories the stem cells in the SAM by grouping all the STM+ positive cells in the atlas (Figure 7F).
- 2) Line 117, The rationale behind using snRNA-seq instead of scRNA-seq is not described. Is it due to a difficulty in preparing high-quality protoplasts for the floral tissues? In addition, there are several papers showing the power of snRNA-seq in plants (e.g. maize atlas recently published in Cell). These papers should be cited.
- 3) Please also cite recent very nice review papers from Birnbaum, Bert De Rybel and Giacomello labs (TIPS, 2018; Annu Rev Plant Biol, 2021; COPB, 2021).
- 4) Methods. I wonder when the floral tissues were harvested after DEX treatment?

Reviewer #3 (Remarks to the Author):

Global gene expression analysis of different cell types in a multicellular context is important to identify gene networks regulating the specification and maintenance of different cell types. This requires the identification of gene expression programs of cells in particular locations and at times even before the manifestation of the morphological changes. The paper mainly focuses on testing an adaptation of the novoSpaRc method in concert with a spatial map in order to accurately predict the spatial expression of genes in developing floral meristems. The earlier developed pAP1:AP1-GR ap1-1 cal-1 system is used to induce flower development and extract nuclei from a single time point at 4 days after Dex application. This single nuclei expression data was later compared with the data obtained from fluorescent reporters Ap3 and Ag which provide spatial reference to generate what is termed 3D expression map. The method can trace the initiation of vascular identity within the floral meristem which in itself is not new as previous single cell gene expression analysis studies have shown such potential.

There are a number of suggestions that could be implemented to more completely round out the study and showcase the power of the approach.

- 1) It is a methodology paper, however, new biological insights on the temporal control of flower development could be obtained if authors extend their analysis to additional time points after ap1 induction which is currently limited to one time point. The assessment of temporal changes in gene expression programs and reorganization or appearance of new gene expression clusters will also help also in the validation of the method which is currently restricted to two markers at a single time point.
- 2) The power of the method can also be demonstrated if authors can find new and more refined expression domains within the already known expression domains. RNA in situ analysis of a few candidate genes within the gene clusters identified is necessary.
- 3) From the technical point of view, there could be more side-by-side comparisons to previous methods or more in-depth discussion of the advantages and drawbacks of previous methods such as

the ones mentioned within the introduction itself. Most of the comparisons present are to experimental results and it is not as clear as it possibly could be what this method brings to the table compared to other computational methods and over the original novoSparRc. Some of the concepts are not detailed in optimal depth for nonspecialists. For example, the nature of the spatial map used with novoSparRc or in what way it differs if any from maps used in previous methods such as DistMap. Some of the figures could use more explanatory legends. Overall as a computational method focused study there should be more detail and data added supporting the uniqueness and advantages of this method. The following suggestions may highlight the uniqueness of the study from the technical standpoint.

- a) How specifically does the 2017 DistMap improve upon the resolution of the 2019 study? Are there alternatives to this package? Or is this a first in kind algorithm?
- b) Fig1: Authors should consider a legend for the intensity of read counts in 1a. 1B: individual clusters are well discussed but what might be the significance of collections of adjacent clusters? 1C: more elaboration of results in 1C is needed. Authors mention about one or two clusters of interest like 0, 10, 11 but other areas of the figure3 are not mentioned. For example, is there any significance to the generally lower expression on the left hand side of the figure?
- c) It is not completely clear how the figure caption and the 1D match up with each with each other. The caption says top 20 marker genes, 7 are shown on the x axes. Or does this figure represent something different than the expression of 7 genes among 12 clusters?
- d) The filtering technique is completely computational and appears to rely on the assumption that greater distance means not present on the spatial map. Is this assumption true enough for this to be a suitable strategy for general use? Authors discuss filtering throughout the paper, maybe careful selection of gene subsets might help in more accurately distinguishing structures.
- e) Authors kept both distance measures but appear to believe Jaccard is superior. Is there any advantage to Hamming distance?
- f) Fig. 3: The expression patterns in the center seem to match up pretty well but at the periphery there appears to be some difference between the reconstructions and the confocal images. Also, what would be the units if any for the red/yellow bar?
- g) Fig 4; Provide brief explanation of Spearman's Rho. Provide some sort of legend to quantify color of graphs. Some of the axes seem inconsistently labeled. For example, 4D has 2.8 at the beginning but it's not clear whether the same applies to 4C.
- h) If a 3D map is necessary for good reconstruction, what would be the advantage over earlier methods mentioned such as DistMap?
- i) Line 367: The reference to Supplemental Figure 11 doesn't seem to match up with the figure?
- j) Line 370; By specific subset do you mean along the central axis of the meristem?

Reviewer #1 (Remarks to the Author):

I am generally positive on this manuscript because authors demonstrated how they used novoSpaRc to construct a 3D expression atlas of Arabidopsis floral meristem, where the transitional unsupervised clustering approach is failed due to the dominance of the epidermal/vascular tissues and dividing cells. This strategy will be useful and of broad interest to the researches working on scRNA-seq in plants. Moreover, authors have generated a webserver which facilitates the use of the datasets generated in this study. However, the paper can be improved in the following aspects:

We thank the reviewer for her/his positive comment.

1) Line 175, Authors' stated the drawback of unsupervised clustering approach. However, it should be noted that several scRNA-seq analyses in Arabidopsis roots have successfully reconstructed gene spatial expression map and inferred underlying differentiation trajectories. These papers should be cited and discussed. Please also note that Zhang et al (Dev Cell, 2021) have proposed another approach to mitigate the dominance of certain cell types during cell clustering. They were able to infer differentiation trajectories the stem cells in the SAM by grouping all the STM+ positive cells in the atlas (Figure 7F).

We thank the reviewer for this comment. Indeed, we agree with the reviewer that unsupervised clustering has been successfully used in the field. However, the outcome of unsupervised clustering is not always certain to identify clusters with particular biological relevance to the user. For example the unsupervised clustering of our snRNA-seq data didn't identify the classical flower meristem expression-domains (AG-, AP3-, and AP1-) . Now, we have modified our manuscript to : "Unsupervised clustering has been successfully used for the analysis of scRNA-seq, however, one of the major drawbacks of this approach is that it identifies groups of cells depending on their transcriptome variance [...]"

2) Line 117, The rationale behind using snRNA-seq instead of scRNA-seq is not described. Is it due to a difficulty in preparing high-quality protoplasts for the floral tissues? In addition, there are several papers showing the power of snRNA-seq in plants (e.g. maize atlas recently published in Cell). These papers should be cited.

We thank the reviewer for his/her comment. It is reported in the literature that the protoplast procedure can lead to changes in gene expression. Now, we have added to our ms: "We isolate nuclei instead of protoplasts to avoid the transcriptome changes that protoplasting may

produce (e.g. Sunaga-Franze et al. 2021; Thibivilliers, Anderson, and Libault 2020; Shulze et al. 2019)".

Marand et al. (2021) and other papers related to snRNA-seq in plants has been now referenced.

3) Please also cite recent very nice review papers from Birnbaum, Bert De Rybel and Giacomello labs (TIPS, 2018; Annu Rev Plant Biol, 2021; COPB, 2021).

Those are indeed very interesting papers and they are now referenced in the introduction.

4) Methods. I wonder when the floral tissues were harvested after DEX treatment?

The tissues were harvested after 4 days of DEX-induction. For this revision, we have processed a second time-point after 3 days of DEX-induction. Now, we say in M&M: "Around 20 inflorescences were collected and snap-frozen in liquid nitrogen on day 3 and day 4 after the first DEX-induction for snRNA-seq and on day 4 after induction for the domain-specific RNA-seq library preparation."

Reviewer #3 (Remarks to the Author):

Global gene expression analysis of different cell types in a multicellular context is important to identify gene networks regulating the specification and maintenance of different cell types. This requires the identification of gene expression programs of cells in particular locations and at times even before the manifestation of the morphological changes. The paper mainly focuses on testing an adaptation of the novoSparRc method in concert with a spatial map in order to accurately predict the spatial expression of genes in developing floral meristems. The earlier developed pAP1:AP1-GR ap1-1 cal-1 system is used to induce flower development and extract nuclei from a single time point at 4 days after Dex application. This Single nuclei expression data was later compared with the data obtained from fluorescent reporters Ap3 and Ag which provide spatial reference to generate what is termed 3D expression map. The method can trace the initiation of vascular identity within the floral meristem which in itself is not new as previous single cell gene expression analysis studies have shown such potential.

We appreciate the nice and accurate summary of the reviewer. However, we must disagree with the comment "[...] initiation of vascular identity within the floral meristem which in itself is not new as previous single cell gene expression analysis studies have shown such potential". We are not aware of any publication describing the expression of vascular genes at the early

stage of flower development (stage 4-5 just when the domains that will rise to the floral organs are established) described in our publication. Our previous work on scRNA-seq in flower tissues (Sunaga-Franze *et al.* 2021) identified cells with expression corresponding to vascular tissues but this was in mature flower stages (just before anthesis). Zhang, Chen, and Wang (2021) describe vascular tissues at the vegetative shoot apex, however, the material used includes not only the SAM, but also leaf primordia at different stages (including more mature tissues).

One of the problems of single-cell methodologies is that we don't know the exact physical location of each of the transcriptomes obtained. Using snRNA-seq (our Fig 1C) we were able to identify cells with vasculature-related transcriptomes. However, until we mapped these transcriptomes to the 3D reconstructed meristem, we always thought that this was some contamination from the vascular tissues of pedicels. Using the 3D reconstruction, we were not only able to identify transcriptomes of cells representing early vasculature differentiation stages, but also we are able to locate them within the flower meristem. In addition, this new finding was validated by gene reporter assays (Sup Fig 15).

There are a number of suggestions that could be implemented to more completely round out the study and showcase the power of the approach.

1) It is a methodology paper, however, new biological insights on the temporal control of flower development could be obtained if authors extend their analysis to additional time points after ap1 induction which is currently limited to one time point. The assessment of temporal changes in gene expression programs and reorganization or appearance of new gene expression clusters will also help also in the validation of the method which is currently restricted to two markers at a single time point.

We thank the reviewer for this comment. Indeed, our long-term research goal is to be able to obtain spatial gene expression profiles at single-cell resolution in the flower meristem at different time points/growth conditions. Our method focuses on mapping scRNA-seq to a 3D microscope-based reconstructed meristem, but to link 3D reconstructed meristems of different stages with different morphologies is yet an unresolved problem and beyond the scope of this paper.

However, we agree with the reviewer that showing some temporal comparison will exemplify the potential of our method. Therefore, we have generated a second snRNA-seq sample at day 3 after DEX-induction, just one day earlier than our previous sample. This allows us to map both

snRNA-seq samples to the same 3D meristem (stage 4-5), and avoid the unsolved problem of aligning 3D meristems from different developmental time-points. Now we have updated our manuscript with a new section called: **“Identifying floral meristem expression-domains and their temporal dynamics”**.

2) The power of the method can also be demonstrated if authors can find new and more refined expression domains within the already known expression domains. RNA in situ analysis of a few candidate genes within the gene clusters identified is necessary.

We thank the reviewer for the comment, which has improved our manuscript.

In order to find new or more refined expression domains within the flower meristem, we cluster the predicted expression profiles of the cells of the reconstructed 3D flower meristem. We identified 15 expression domains, 7 of them in the L1 layer. Refahi *et al.* recently identified 11 domains in the L1 layer of the flower meristem. Our expression domains have an almost perfect correspondence with Refahi’s expression domains (see new Sup Fig. 8), which supports the predictive power of our method to identify known expression-domains. The domains identified by Refahi *et al.* were already validated by using a large set of RNA *in situ* and reporter genes experiments, therefore we believe that no additional RNA *in situ* hybridization or reporter gene analysis is needed to validate these domains.

In addition, our approach identified several new domains in the bottom of the meristem, which mainly correspond to vascular tissues. These approximate positions of the vasculature domains were validated using new reporter gene experiments for specific vascular marker genes (PXY, and SMXL5 Sup Fig 15), and by comparing with the RNA-seq expression of cells expressing PXY or SMXL5 (Sup Fig 16).

Accordingly, we added a new section to the manuscript called: **“Identifying floral meristem expression-domains and their temporal dynamics”**.

3) From the technical point of view, there could be more side-by-side comparisons to previous methods or more in-depth discussion of the advantages and drawbacks of previous methods such as the ones mentioned within the introduction itself. Most of the comparisons present are to experimental results and it is not as clear as it possibly could be what this method brings to the table compared to other computational methods and over the original novoSparRc. Some of the concepts are not detailed in optimal depth for nonspecialists. For example, the nature of the spatial map used with novoSparRc or in what way it differs if any from maps used in previous methods such as DistMap. Some of the figures could use more explanatory legends. Overall as a computational method focused study there should be more detail and data added

supporting the uniqueness and advantages of this method. The following suggestions may highlight the uniqueness of the study from the technical standpoint.

One unique feature of novoSpaRc is that it requires only a small number of reference genes to map the scRNA-seq data to the spatial map, while other computational methods require hundreds or thousands of reference genes. In the plant field, spatial maps are generated using confocal microscopy of fluorescent reporter genes or RNA *in situ* hybridization, resulting in spatial maps with a very limited number of reference genes (in our case, 23 reference genes). Therefore, novoSpaRc is the unique method to be used with this low number of reference genes. Now, we have updated the introduction and discussion to make this point more clear. In addition, we reference the method of Bravo González-Blas et al. (2020), which does not require reference genes but main axis (e.g. ventral-dorsal) of expression variation in the organs studied. This method is now compared and discussed in the discussion section: “Recently, Bravo González-Blas et al. (2020) implemented a new methodology to map scRNA-seq and/or scATAC-seq to a 1D or 2D spatial map. The method doesn’t require any reference gene and it is based into: 1) Link the expression domains already know in the spatial map to the cell clusters identified and annotated in the scRNA-seq (e.g: cells from the cluster annotated to epidermis will be linked to the epidermis regions of the spatial map). 2) To use pseudo-time analysis to order the transcriptomes of the scRNA-seq in a spatial axis, this assumes that there is only one axis of variation in the spatial expression patterns (e.g. ventral-dorsal axis), and the expression patterns are symmetric around the axis. Although, we can imagine that this will be a valid approach to some root tissues, where one main axis of expression variation is present (from root cap to maturation region), it is not the case for the complex 3D morphology of the flower meristem.”

a) How specifically does the 2017 DistMap improve upon the resolution of the 2019 study? Are there alternatives to this package? Or is this a first in kind algorithm?

Our reference to Stuart et al. (2019) was incorrect, it should have been Satija et al. (2015). This paper together with Achim et al (2015) were the first ones to implement the idea of mapping scRNA-seq to a spatial map. Now, we also reference alternative methods.

DistMap improves the resolution because the previous methods computationally bin the cells in regions while DistMap treats each cell individually.

There are other alternatives to novoSpaRc, but they require a larger number of reference genes in the spatial map, which is very rare or not existent in the plant field. Now we have improved our introduction with the paragraph: “Mapping of scRNA-seq transcriptomes into a

computational representation of the studied organ/structure provides an alternative method for spatial reconstruction of omics data. Two seminal papers implemented this idea by mapping scRNA-seq data to a computationally binned spatial map consisting in the expression of ~100 reference genes (Satija et al. 2015; Achim et al. 2015). This idea, with different implementations, was successfully followed by others in diverse tissues and organisms (Halpern et al. 2017; Waldhaus, Durruthy-Durruthy, and Heller 2015; Durruthy-Durruthy et al. 2014; Moor et al. 2018; Naomi et al. 2016; Karaiskos et al. 2017). New methods aim to combine scRNA-seq with high-throughput spatial transcriptome data (e.g. MERFISH, Slide-seq) that collect the expression of thousands of reference genes. They are based on the projection of the scRNA-seq and the spatial transcriptomes into a common latent space e.g. SEURAT (Stuart et al. 2019), Liger (Welch et al. 2019), Harmony (Korsunsky et al. 2019), gimVI (Lopez et al. 2018), SpaGe (Abdelaal et al. 2020). In general, there is a tendency to develop computational methods that require a large number of reference genes, which limits these tools to organisms with extensive spatial transcriptomics resources.”

b) Fig1: Authors should consider a legend for the intensity of read counts in 1a.

We agree. A legend has been added.

1B: individual clusters are well discussed but what might be the significance of collections of adjacent clusters?

Adjacent clusters in Fig 1B should have similar transcriptomes. This could be because they represent similar tissue types (e.g. epidermis or vasculature), cells in similar cell cycle stage/s (i.e. dividing cells), cells coming from the same lineage, or, most likely, a mix of all these factors. As we work with synchronized tissues, it is unlikely that the proximity of the clusters to each other represents some type of temporal relationship. Now we have added to the text: “The clusters appear to be grouped by the tissue where they are located (epidermis versus vasculature and parenchyma), and their cell cycle status”

A deeper discussion about this is interesting, but we believe this is out of the scope of our methodological paper where we want to show the potential of mapping the snRNA-seq data to a reconstructed meristem.

1C: more elaboration of results in 1C is needed. Authors mention about one or two clusters of interest like 0, 10, 11 but other areas of the figure3 are not mentioned. For example, is there any significance to the generally lower expression on the left hand side of the figure?

In the manuscript, we described all the identified clusters (lines 139 and 174). We started describing the epidermal clusters (0, 9, 10, and 11), in the next paragraph, we describe the vasculature clusters (1, 8, 12), and finally clusters 2, 3, 5, and 6.

Fig. 1C shows the average relative expression of known floral markers in the identified clusters as described in Material & Methods. Now, we have updated the legend of Fig. 1C to make this clear. When a gene shows low relative expression means that several clusters have similar expression values, and the gene is not specific to any particular cluster. For better visualization, we cluster the values of the heatmap, in this way, genes with similar relative expression patterns cluster together.

c) It is not completely clear how the figure caption and the 1D match up with each with each other. The caption says top 20 marker genes, 7 are shown on the x axes. Or does this figure represent something different than the expression of 7 genes among 12 clusters?

Indeed, the legend was unclear. Now, we have modified the legend to say: *“Relationship between domain-specific shoot apical meristem bulk RNA-seq datasets profiled by (Tian et al. 2019) and the snRNA-seq clusters. The heatmap shows the relative average expression of the top 20 marker genes for each snRNA-seq cluster (y-axis) on domain-specific shoot apical meristem bulk RNA-seq datasets. For example, the top 20 marker genes of cluster 1 have high specific expression on the ATHB8-domain, meaning that they are specific to this domain.”*

d) The filtering technique is completely computational and appears to rely on the assumption that greater distance means not present on the spatial map. Is this assumption true enough for this to be a suitable strategy for general use? Authors discuss filtering throughout the paper, maybe careful selection of gene subsets might help in more accurately distinguishing structures.

We thank the reviewer for this comment. The filtering step is introduced to eliminate cells from regions present in the snRNA-seq material and not in the spatial map (e.g. non-floral cells from short pedicels and stems, or some type of contamination). When these regions are *a priori* known, and a large number of specific markers for these regions are also known, one could envision a method to identify and eliminate these cells using a careful selection of marker genes. However, a more objective approach is to set a filtering based on the transcriptome dissimilarity to the spatial map which doesn't require further assumptions about the nature of the contamination.

e) Authors kept both distance measures but appear to believe Jaccard is superior. Is there any advantage to Hamming distance?

For binary data, the most common distances that can be calculated are the Hamming and *Jaccard* distance. We didn't *a priori* know which distance had better performance for our particular dataset, so we compared both and also the *euclidean* distance (see sup Fig 5). For the dataset presented in the manuscript, the best performance distance was *Jaccard* (see sup Fig 5), and all further analyses were obtained using only the *Jaccard* distance. We informed the reader that other datasets might have different optimal parameter settings.

f) Fig. 3: The expression patterns in the center seem to match up pretty well but at the periphery there appears to be some difference between the reconstructions and the confocal images.

The accuracy of our method depends on the number of reference genes available to link the snRNA-seq and microscopy data. Indeed, we have fewer reference genes in the periphery than in the center. This could cause some differences. Therefore, we have updated our manuscript saying: "In general, the prediction broadly recovered the cells and tissues that show activities of the genes, but some gene expression patterns were more restricted in the reporter gene analyses (e.g. *SHR*, *PIN1*). This could be explained by the limited set of reference genes that were used for the prediction, in particular in the periphery where fewer reference genes were available [...]"

Also, what would be the units if any for the red/yellow bar?

We thank the reviewer for noticing this omission. The expression values were standardized to have mean 0 and variance 1, therefore they have no units. Now we have added to the legend: "To facilitate visual comparison, we standardized the expression of each gene to have mean 0 and variance 1."

g) Fig 4; Provide brief explanation of Spearman's Rho. Provide some sort of legend to quantify color of graphs. Some of the axes seem inconsistently labeled. For example, 4D has 2.8 at the beginning but it's not clear whether the same applies to 4C.

We thank the reviewer for pointing out this. Now, we have included in the legend the following text: "On the top of each figure is reported the Spearman' rank correlation coefficient (Rho)".

Now a legend has been included and the scale was corrected.

h) If a 3D map is necessary for good reconstruction, what would be the advantage over earlier methods mentioned such as DistMap?

Early methods (and most of the new methods) map scRNA-seq data to 2D representations of particular organs or structures. This has the clear disadvantage to not be able to reconstruct the 3D organ/structure, ignoring the transcriptome complexity that can happen in other layers of the organ/tissue.

i) Line 367: The reference to Supplemental Figure 11 doesn't seem to match up with the figure?

Yes, the reference should be to sup Figure 12. Now, we have corrected this.

j) Line 370; By specific subset do you mean along the central axis of the meristem?

Yes, now we say: "[...] in a specific subset of cells along the central axis of the meristem."

REFERENCES

- Achim, Kaia, Jean-Baptiste Pettit, Luis R Saraiva, Daria Gavriouchkina, Tomas Larsson, Detlev Arendt, and John C Marioni. 2015. "High-Throughput Spatial Mapping of Single-Cell RNA-Seq Data to Tissue of Origin." *Nature Biotechnology* 33 (5): 503–9. <https://doi.org/10.1038/nbt.3209>.
- Bravo González-Blas, Carmen, Xiao-Jiang Quan, Ramon Duran-Romaña, Ibrahim Ihsan Taskiran, Duygu Koldere, Kristofer Davie, Valerie Christiaens, et al. 2020. "Identification of Genomic Enhancers through Spatial Integration of Single-Cell Transcriptomics and Epigenomics." *Molecular Systems Biology* 16 (5): e9438. <https://doi.org/https://doi.org/10.15252/msb.20209438>.
- Bravo González-Blas, Carmen, Xiao-Jiang Quan, Ramon Duran-Romaña, Ibrahim Ihsan Taskiran, Duygu Koldere, Kristofer Davie, Valerie Christiaens, et al. 2020. "Identification of Genomic Enhancers through Spatial Integration of Single-cell Transcriptomics and Epigenomics." *Molecular Systems Biology* 16 (5). <https://doi.org/10.15252/msb.20209438>.
- Durruthy-Durruthy, Robert, Assaf Gottlieb, Byron H Hartman, Jörg Waldhaus, Roman D Laske, Russ Altman, and Stefan Heller. 2014. "Reconstruction of the Mouse Otocyst and Early Neuroblast Lineage at Single-Cell Resolution." *Cell* 157 (4): 964–78. <https://doi.org/10.1016/j.cell.2014.03.036>.
- Halpern, Keren Bahar, Rom Shenhav, Orit Matcovitch-Natan, Beáta Tóth, Doron Lemze, Matan Golan, Efi E Massasa, et al. 2017. "Single-Cell Spatial Reconstruction Reveals Global Division of Labour in the Mammalian Liver." *Nature* 542 (7641): 352–56. <https://doi.org/10.1038/nature21065>.
- Karaiskos, Nikos, Philipp Wahle, Jonathan Alles, Anastasiya Boltengagen, Salah Ayoub, Claudia

- Kipar, Christine Kocks, Nikolaus Rajewsky, and Robert P Zinzen. 2017. "The &Drosophila& Embryo at Single-Cell Transcriptome Resolution." *Science* 358 (6360): 194 LP – 199. <https://doi.org/10.1126/science.aan3235>.
- Marand, Alexandre P, Zongliang Chen, Andrea Gallavotti, and Robert J Schmitz. 2021. "A cis-Regulatory Atlas in Maize at Single-Cell Resolution." *Cell* 184 (11): 3041-3055.e21. <https://doi.org/10.1016/j.cell.2021.04.014>.
- Moor, Andreas E, Yotam Harnik, Shani Ben-Moshe, Efi E Massasa, Milena Rozenberg, Raya Eilam, Keren Bahar Halpern, and Shalev Itzkovitz. 2018. "Spatial Reconstruction of Single Enterocytes Uncovers Broad Zonation along the Intestinal Villus Axis." *Cell* 175 (4): 1156-1167.e15. <https://doi.org/https://doi.org/10.1016/j.cell.2018.08.063>.
- Naomi, Habib, Li Yinqing, Heidenreich Matthias, Swiech Lukasz, Avraham-Davidi Inbal, Trombetta John J., Hession Cynthia, Zhang Feng, and Regev Aviv. 2016. "Div-Seq: Single-Nucleus RNA-Seq Reveals Dynamics of Rare Adult Newborn Neurons." *Science* 353 (6302): 925–28. <https://doi.org/10.1126/science.aad7038>.
- Refahi, Yassin, Argyris Zardilis, Gaël Michelin, Raymond Wightman, Bruno Leggio, Jonathan Legrand, Emmanuel Faure, et al. 2021. "A Multiscale Analysis of Early Flower Development in Arabidopsis Provides an Integrated View of Molecular Regulation and Growth Control." *Developmental Cell* 56 (4): 540-556.e8. <https://doi.org/10.1016/j.devcel.2021.01.019>.
- Satija, Rahul, Jeffrey A Farrell, David Gennert, Alexander F Schier, and Aviv Regev. 2015. "Spatial Reconstruction of Single-Cell Gene Expression Data." *Nature Biotechnology* 33 (5): 495–502. <https://doi.org/10.1038/nbt.3192>.
- Shulze, Christine N, Benjamin J Cole, Doina Ciobanu, Junyan Lin, Yuko Yoshinaga, Mona Gouran, Gina M Turco, et al. 2019. "High-Throughput Single-Cell Transcriptome Profiling of Plant Cell Types." *Cell Reports* 27 (7): 2241-2247.e4. <https://doi.org/https://doi.org/10.1016/j.celrep.2019.04.054>.
- Stuart, Tim, Andrew Butler, Paul Hoffman, Christoph Hafemeister, Eftymia Papalexi, William M Mauck III, Yuhao Hao, Marlon Stoeckius, Peter Smibert, and Rahul Satija. 2019. "Comprehensive Integration of Single-Cell Data." *Cell* 177 (7): 1888-1902.e21. <https://doi.org/10.1016/j.cell.2019.05.031>.
- Sunaga-Franze, Daniele Y, Jose M Muino, Caroline Braeuning, Xiaocai Xu, Minglei Zong, Cezary Smaczniak, Wenhao Yan, et al. 2021. "Single-Nuclei RNA-Sequencing of Plant Tissues." *BioRxiv*, January, 2020.11.14.382812. <https://doi.org/10.1101/2020.11.14.382812>.
- Thibivilliers, Sandra, Dirk Anderson, and Marc Libault. 2020. "Isolation of Plant Root Nuclei for Single Cell RNA Sequencing." *Current Protocols in Plant Biology* 5 (4): e20120. <https://doi.org/https://doi.org/10.1002/cppb.20120>.
- Waldhaus, Jörg, Robert Durruthy-Durruthy, and Stefan Heller. 2015. "Quantitative High-Resolution Cellular Map of the Organ of Corti." *Cell Reports* 11 (9): 1385–99. <https://doi.org/10.1016/j.celrep.2015.04.062>.
- Zhang, Tian-Qi, Yu Chen, and Jia-Wei Wang. 2021. "A Single-Cell Analysis of the Arabidopsis Vegetative Shoot Apex." *Developmental Cell* 56 (7): 1056-1074.e8. <https://doi.org/https://doi.org/10.1016/j.devcel.2021.02.021>.

REVIEWER COMMENTS

Reviewer #3 (Remarks to the Author):

Authors have performed new experiments and new analyses which exemplify the possible utility of the work presented. The new results on the temporal changes in gene expression reveal early gene expression changes that occur even before morphological changes associated with flower development. The changes that have been made with regard to the figure presentation and legends have improved the clarity of the manuscript.

Reviewer #4

(see attached)

In the manuscript, Neumann et al. utilized 10x Visium platform to analyze the spatial snRNA heterogeneity in the plants based on novoSpaRc. As a proof-of-concept investigation, the concept is novel. Overall, the experiment pipeline and manuscript are well-written but there are some concerns in terms of statistical analysis and methodology. Only if the authors properly address the raised concerns, the manuscript will be considered for publications.

Major revision:

1. Process of the snRNA-seq data. The authors use the Seurat for clustering with 3000 highly variable genes (HVGs). By convention, people use top 2000 HVGs, is there a reason for adopting 3000 HVGs?
2. Similar to the point 1, why adopt 100 highest correlated genes or the top 1000 for calculating the distances (line 218). What is the definition of the 28 reference genes (line 206 and 605)? The authors need to elaborate the step of selecting the 100 highly correlated genes in snRNA-seq data, specifically, correlation calculations in line 674-677 are confusing.
3. In the line 189-192, the authors talked about the missing cell type from the clustering due to the small inter-cluster variance. However, personally, I think it is because the selected marker genes or HVGs for clustering. There is an unsupervised method called FEAST¹ which allows to select the most representative features (genes) before clustering, which is demonstrated better than Seurat for selecting highly variable genes. The authors should try this method and discuss the potential benefits from marker gene selections.
4. The authors adopt different approaches for distance calculations (Supplementary Figure S5), the authors should discuss the result why using Jaccard similarity enjoys the best performance.
5. The authors use AUROC and PEP for assessment. They need to clarify how to calculate the AUROC for each gene based on gene expression values? Personally, AUROC is based on classification predictions not on continuous values. For using PEP as an evaluation, I am concerned that the predicted gene expressions only highly correlated with several genes from the reference gene list. If this was the case, I would doubt the predicted power because it is not generalizable.

Minor revision:

1. Figure1A, the x label should be (snRNA-seq)?

Reference:

1. Su, K., Yu, T. & Wu, H. Accurate feature selection improves single-cell RNA-seq cell clustering. *Briefings in Bioinformatics* (2021) doi:10.1093/bib/bbab034.

Dear Reviewer,

Herewith we submit the revised version of our manuscript for consideration. We would like to thank you for the time invested and for your insightful revision of our manuscript. We have addressed your comments below. All changes in the manuscript are coded in red.

We hope that those changes improve the clarity of our manuscript and that this work is a valuable resource for the plant community.

With kind regards, on behalf of all authors,
Jose Muino

Reviewer #4 (Remarks to the Author):

In the manuscript, Neumann et al. utilized 10x Visium platform to analyze the spatial snRNA heterogeneity in the plants based on novoSpaRc. As a proof-of-concept investigation, the concept is novel. Overall, the experiment pipeline and manuscript are well-written but there are some concerns in terms of statistical analysis and methodology. Only if the authors properly address the raised concerns, the manuscript will be considered for publications.

Major revision:

1. Process of the snRNA-seq data. The authors use the Seurat for clustering with 3000 highly variable genes (HVGs). By convention, people use top 2000 HVGs, is there a reason for adopting 3000 HVGs?

We thank the reviewer for the time taken to go through our M&M. As already indicated in the paper, we used the *SCTransform* function within the SEURAT package to normalize the data. The default value of this function is to select the top 3,000 HVGs (<https://rdrr.io/github/satijalab/seurat/man/SCTransform.html>). Recent papers in the plant field also used this parameter value (eg. Sahan et. al [1]).

The use of top 2,000 HVGs is the default value of the function *FindVariableFeatures* which is typically used with a previous type of normalization (*ScaleData*) within the SEURAT package. Because two functions in the SEURAT package used different default values for this parameter (top HGVs), we explicitly stated the value used in the paper. Now, we have modified the text to: "SCT-normalization was applied within the SEURAT package with default values" to avoid confusion.

1. Similar to the point 1, why adopt 100 highest correlated genes or the top 1000 for calculating the distances (line 218). What is the definition of the 28 reference genes (line 206 and 605)? The authors need to elaborate the step of selecting the 100 highly correlated genes in snRNA-seq data, specifically, correlation calculations in line 674-677 are confusing.

The term "reference genes" was defined in line 204. Now, we have improved the definition to: "a published 3D reconstructed *Arabidopsis* stage 4-5 floral meristem ("spatial map") that has information on the expression pattern of 28 well-known marker genes of different floral meristem regions ("reference genes") (Refahi et al. 2021) [2]."

We used the top 100 correlated genes because this leads to the selection of 1,709 genes to generate a low dimensional representation of the data. 2,000-3,000 is the typical number of genes used by SEURAT and other similar programs to generate this representation (see previous answer). When using the top 1,000 genes, we will get a selection of 11,407 genes which is much larger than usual.

1. The authors adopt different approaches for distance calculations (Supplementary Figure S5), the authors should discuss the result why using Jaccard similarity enjoys the best performance.

As we have already stated in the paper, the Jaccard distance enjoys the best performance in the dataset studied in this paper, but for other datasets we found that other distances (eg: euclidean) performed better. Now, we have modified our text to make this more clear to the reader: "In particular, using the Jaccard distance had a positive impact on the performance of the method in this particular dataset (Sup Fig. 5). In our hands, other datasets show different optimal parameter settings, but filtering always improves the performance. Potential users of this method are advised to find the optimal settings for their specific datasets of interest."

1. The authors use AUROC and PEP for assessment. They need to clarify how to calculate the AUROC for each gene based on gene expression values? Personally, AUROC is based on classification predictions not on continuous values. For using PEP as an evaluation, I am concerned that the predicted gene expressions only highly correlated with several genes from the reference gene list. If this was the case, I would doubt the predicted power because it is not generalizable.

Indeed, the AUROC assessment was not clear. We have now added to the text: "In particular, the AUROC for each gene was calculated using the ROCR package (Sing et al. 2005) [4], where the predicted expression values serve as a score to indicate expression or not expression in each cell of the spatial map."

If we correctly understood the reviewer, he/she is concerned that the PEP score calculated with our specific reference set of genes will not be generalizable for other spatial expression maps or when using other reference genes. Indeed, we agree with the reviewer. Users of our method should recalculate the PEP score when the set of reference genes or the tissue under investigation changes. Now, we have modified our text to: "Thus, we provide a method (PEP score) that can be used to estimate the performance of the predicted expression for each gene given a previously defined set of reference genes, even before having generated the reference spatial map."

Minor revision:

1. Figure 1A, the x label should be (snRNA-seq)?

Thank you for pointing out this mistake. The label on the x-axis in Figure 1A has been corrected.

References:

[1] Shahan, R., Hsu, C.-W., Nolan, T. M., Cole, B. J., Taylor, I. W., Greenstreet, L., Zhang, S., Afanassiev, A., Vlot, A. H. C., Schiebinger, G., Benfey, P. N., & Ohler, U. (2022). A single-cell Arabidopsis root atlas reveals developmental trajectories in wild-type and cell identity mutants. *Developmental Cell*.
<https://doi.org/https://doi.org/10.1016/j.devcel.2022.01.008>

[2] Refahi, Y., Zardilis, A., Michelin, G., Wightman, R., Leggio, B., Legrand, J., Faure, E., Vachez, L., Armezzani, A., Risson, A.-E., Zhao, F., Das, P., Prunet, N., Meyerowitz, E. M., Godin, C., Malandain, G., Jönsson, H., & Traas, J. (2021). A multiscale analysis of early flower development in Arabidopsis provides an integrated view of molecular regulation and growth control. *Developmental Cell*, 56(4), 540-556.e8.
<https://doi.org/10.1016/j.devcel.2021.01.019>

- [3] Su, K., Yu, T., & Wu, H. (2021). Accurate feature selection improves single-cell RNA-seq cell clustering. *Briefings in Bioinformatics*, 22(5), bbab034.
<https://doi.org/10.1093/bib/bbab034>
- [4] Sing, T., Sander, O., Beerenwinkel, N., & Lengauer, T. (2005). ROCR: visualizing classifier performance in R. *Bioinformatics*, 21(20), 3940–3941.
<https://doi.org/10.1093/bioinformatics/bti623>

REVIEWERS' COMMENTS

Reviewer #4 (Remarks to the Author):

The authors answer my concern mostly. However, they should consider including the results from FEAST and Seurat together to increase the robustness of the results.

I wonder whether other publications also adopt PEP measurement for evaluations?

Dear Reviewer,

We would like to thank the reviewer for the time invested reviewing the manuscript. We have addressed her/his comments below.

We hope that those changes improve the clarity of our manuscript and that this work is a valuable resource for the plant community.

With kind regards, on behalf of all authors,
Jose Muino

REVIEWERS' COMMENTS

Reviewer #4 (Remarks to the Author):

The authors answer my concern mostly. However, they should consider including the results from FEAST and Seurat together to increase the robustness of the results.

Thanks for the suggestion. We have considered including the results from FEAST, but we have decided not to include them. The main reason is because the focus of our paper is to integrate scRNA-seq datasets with microscopy-based spatial maps. The comparison between FEAST and Seurat is about how to select the set of high variable genes to do the analysis of scRNA-seq, but it is not directly related with the integration of scRNA-seq and spatial maps.

I wonder whether other publications also adopt PEP measurement for evaluations?

We are not aware of any other publication adopting PEP measurements for evaluations. There are very few papers addressing the integration of scRNA-seq data with spatial maps and we believe that we are the first ones to propose a method to predict which spatial genes expression patterns are correctly estimated.